



# Limitations in representation of physical processes prevent successful simulation of PM2.5 during KORUS-AQ

Katherine R. Travis[1], James H. Crawford[1], Gao Chen[1], Carolyn E. Jordan[1,2], Benjamin A. Nault[3], Hwajin Kim[4], Jose L. Jimenez[5], Pedro Campuzano-Jost[5], Jack E. Dibb[6], Jung-Hun Woo[7], Younha Kim[8], Shixian Zhai[9], Xuan Wang[10], Erin E. McDuffie[11], Gan Luo[12], Fangqun Yu[12], Saewung Kim[13], Isobel J. Simpson[14], Donald R. Blake[14], Limseok Chang[15], Michelle J. Kim[16]

[1]NASA Langley Research Center, Hampton, VA, USA
[2]National Institute of Aerospace, Hampton, VA, USA
[3]Center for Aerosol and Cloud Chemistry, Aerodyne Research Inc. 45 Manning Road Billerica, MA, USA
[4]Department of Environmental Health Sciences, Graduate School of Public Health, Seoul National University, Seoul 08826, Korea
[5]Cooperative Institute for Research in the Environmental Sciences, University of Colorado, Boulder, Colorado, USA
[6]Earth System Research Center, University of New Hampshire, Durham, NH, USA
[7]Department of Civil and Environmental Engineering, Konkuk University, Seoul, Republic of Korea
[8]Energy, Climate, and Environment (ECE) Program, International Institute for Applied Systems Analysis (IIASA), Laxenburg, Austria
[9]John A. Paulson School of Engineering and Applied Sciences, Harvard University, Cambridge, MA, USA
[10]City University of Hong Kong, Kowloon, HK
[11]Department of Energy, Environmental, and Chemical Engineering, Washington University in St. Louis, St. Louis, MO, USA
[12]Atmospheric Sciences Research Center, University at Albany, Albany, NY, USA
[13]University of California, Irvine, Irvine, CA, USA
[14]Department of Chemistry, University of California, Irvine, California, USA
[15]Air Quality Research Division, National Institute of Environmental Research, Incheon, Republic of Korea
[16]Division of Geological and Planetary Sciences, California Institute of Technology, Pasadena, CA, USA

*Correspondence to*: K. R. Travis, katherine.travis@nasa.gov

## Abstract.

High levels of fine particulate matter (PM2.5) pollution in East Asia often exceed local air quality standards. Observations from the Korea United States-Air Quality (KORUS-AQ) field campaign in May and June 2016 showed that development of extreme pollution (haze) occurred through a combination of long-range transport and favorable meteorological conditions that enhanced local production of PM2.5. Atmospheric models often have difficulty simulating PM2.5 chemical composition during haze, which is of concern for the development of successful control measures. We use observations from KORUS-AQ to examine the ability of the GEOS-Chem chemical transport model to simulate PM2.5 composition throughout the campaign and identify the mechanisms driving the pollution event. In the surface level, the model underestimates campaign average sulfate aerosol by -64% but overestimates nitrate aerosol by 36%. The largest underestimate in sulfate occurs during the pollution event in conditions of high relative humidity, where models typically struggle to generate the high concentrations due to missing



heterogeneous chemistry in aerosol liquid water in the polluted boundary layer. Hourly surface observations show that the model nitrate bias is driven by an overestimation of the nighttime peak. In the model, nitrate formation is limited by the supply of nitric acid, which is biased by +100% against aircraft observations. We hypothesize that this is due to a missing sink, which we implement here as a factor of five increase in dry deposition. We show that the resulting increased deposition velocity is consistent with observations of total nitrate as a function of photochemical age. The model does not account for factors such as the urban heat island effect or the heterogeneity of the built-up urban landscape resulting in insufficient model turbulence and surface area over the study area that likely results in insufficient dry deposition. Other species such as $NH_3$ could be similarly affected but were not measured during the campaign. Nighttime production of nitrate is driven by $NO_2$ hydrolysis in the model, while observations show that unexpectedly elevated nighttime ozone (not present in the model) should result in $N_2O_5$ hydrolysis as the primary pathway. The model is unable to represent nighttime ozone due to an overly rapid collapse of the afternoon mixed layer and excessive titration by NO. We attribute this to missing nighttime heating driving deeper nocturnal mixing that would be expected to occur in a city like Seoul. This urban heating is not considered in air quality models run at large enough scales to treat both local chemistry and long-range transport. Key model failures in simulating nitrate, mainly overestimated daytime nitric acid, incorrect representation of nighttime chemistry, and an overly shallow and insufficiently turbulent nighttime mixed layer, exacerbate the model's inability to simulate the buildup of $PM_{2.5}$ during haze pollution. To address the underestimate in sulfate most evident during the haze event, heterogeneous aerosol uptake of $SO_2$ is added to the model which previously only considered aqueous production of sulfate from $SO_2$ in cloud water. Implementing a simple parameterization of this chemistry improves the model abundance of sulfate but degrades the $SO_2$ simulation implying that emissions are underestimated. We find that improving model simulations of sulfate has direct relevance to determining local vs. transboundary contributions to $PM_{2.5}$. During the haze pollution event, the inclusion of heterogeneous aerosol uptake of $SO_2$ decreases the fraction of $PM_{2.5}$ attributable to long-range transport from 66% to 54%. Locally-produced sulfate increased from 1% to 46% of locally-produced $PM_{2.5}$, implying that local emissions controls would have a larger effect than previously thought. However, this additional uptake of $SO_2$ is coupled to the model nitrate prediction which affects the aerosol liquid water abundance and chemistry driving sulfate-nitrate-ammonium partitioning. An additional simulation of the haze pollution with heterogeneous uptake of $SO_2$ to aerosol and simple improvements to the model nitrate simulation results in 30% less sulfate due to 40% less nitrate and aerosol water, and results in an underestimate of sulfate during the haze event. Future studies need to better consider the impact of model physical processes such as dry deposition and boundary layer mixing on the simulation of nitrate and the effect of improved nitrate simulations on the overall simulation of secondary inorganic aerosol (sulfate+nitrate+ammonium) in East Asia. Foreign emissions are rapidly changing, increasing the need to understand the impact of local emissions on $PM_{2.5}$ in South Korea to ensure continued air quality improvements.



## 1 1 Introduction

South Korea enacted legislation in 2018 to address local air pollution, which ranked 13[th] in the world for the worst annual average fine particulate matter (PM$_{2.5}$) exposure levels (Energy Policy Institute, 2019). Ambient PM$_{2.5}$ was the 5[th] highest risk factor for human health in South Korea in 2018, leading to over 20,000 attributable deaths (GBD, 2021). The government plans to reduce the number of days with pollution warnings (PM$_{2.5}$ > 90 µg m$^{-3}$ for two hours) by 50% in 2022 from the 89 that occurred in 2016 (Kim et al., 2018). The reduction of PM$_{2.5}$ levels through policy measures relies on a thorough understanding of pollution sources and the ability of models to simulate potential control measures. Modeling studies have concluded that on average, approximately half of observed PM$_{2.5}$ in South Korea is attributable to long-range transport from China (Lee et al., 2017; Choi et al., 2019; Jung et al., 2019; Kumar et al., 2021). This finding is based on models that have received limited testing of their ability to simulate PM$_{2.5}$ chemical composition, particularly during extreme pollution events. Quantifying the effect of long-range transport relies on regional to global-scale models that trade-off the high resolution needed to resolve urban scales with a large enough domain to represent both the study area and upwind source regions. This evaluation is critical as the contribution of long-range transport to PM$_{2.5}$ in South Korea may be declining due to effective emission controls in China (Han et al., 2021), increasing the need to understand the impact of local emissions on pollution events.

Across East Asia, densely populated regions experience haze events with extremely high levels of PM$_{2.5}$ frequently associated with periods of elevated relative humidity and low daytime mixed layer heights (An et al., 2019). These conditions are favorable for increasing gas-particle partitioning of aerosol precursors. In haze, secondary inorganic aerosol (secondary sulfate+nitrate+ammonium $\equiv$ SNA) is often the dominant component of PM$_{2.5}$, but models have difficulty SNA, particularly sulfate, during these periods (Wang et al., 2014; Zheng et al., 2015a, 2015b; Shao et al., 2019). More generally, the MICS-Asia multi-model comparison showed that the annual contribution of SNA to total PM$_{2.5}$ varied by a factor of two across models and the models also overpredicted the gas-particle partitioning of nitrate (Chen et al., 2019). In the global AeroCom III intercomparison, models differed in their annual concentrations of nitrate and its precursor, nitric acid, by factors of thirteen and nine, respectively (Bian et al., 2017). Models also struggle to represent organic aerosol, overestimating primary organic aerosol (POA) but underestimating secondary organic aerosol (SOA, Zhao et al., 2016), possibly due to missing sources from anthropogenic precursors (Nault et al., 2020). This wide range of model performance in simulating PM$_{2.5}$ composition emphasizes the urgent need to improve our understanding of the sources and conditions driving haze events.

Improving model representation of sulfate chemistry cannot be considered entirely separately from model nitrate biases. In the atmosphere, aqueous-phase chemistry is a major source of sulfate, where clouds provide the dominant source of liquid water (Herrmann et al., 2015). Recent studies have hypothesized that the high aerosol liquid water content (ALWC) associated with PM$_{2.5}$ during extreme pollution events in East Asia allows for significant sulfate production not considered in most models (Wang et al., 2014; Zheng et al., 2015a, 2015b; Shao et al., 2019). Levels of ALWC are very sensitive to aerosol nitrate (Ge





et al., 2012; Sun et al., 2018). The aqueous pathway(s) for $SO_2$ oxidation in ALWC are uncertain in part due to poor understanding of aerosol acidity (An et al., 2019), a key factor controlling nitric acid - nitrate partitioning (Guo et al., 2016). Accurate simulation of aerosol composition and water uptake is required to interpret satellite observations of aerosol optical depth (AOD) (Saide et al., 2020) and evaluate the reponse to emission changes.

05     The Korea United States-Air Quality campaign (KORUS-AQ), conducted in May and June 2016 in South Korea (Crawford et al., 2021), provides an extensive set of ground and aircraft-based observations that can further constrain model simulations of the chemical and physical drivers of $PM_{2.5}$. The campaign included a haze event with concentrations exceeding local air quality standards, characterized by rapid buildup of SNA aerosol. Throughout KORUS-AQ, surprisingly high levels of nighttime ozone, particularly prevalent during haze, appeared to drive nighttime nitrate formation through $N_2O_5$ hydrolysis (Jordan et

al., 2020). This was attributed to elevated nocturnal mixed layer heights (MLH). Zhai et al. (2021) found a severe model overestimate in nighttime nitrate during KORUS-AQ, implying a failure to correctly simulate these conditions. We use the GEOS-Chem chemical transport model applied at high resolution (0.25º × 0.3125º) over East Asia to investigate model representation of $PM_{2.5}$ mass and chemical composition during KORUS-AQ. We specifically evaluate model performance during the conditions governing the development of haze pollution such as elevated relative humidity, increased SNA, and

high nighttime ozone levels. We demonstrate how addressing deficiencies in model physical processes (e.g., nighttime mixing, deposition) are fundamental to the successful simulation of $PM_{2.5}$.

## 2 KORUS-AQ observations

The KORUS-AQ campaign (Crawford et al., 2021) was a joint field campaign organized by South Korea's National Institute of Environmental Research (NIER) and the United States National Aeronautics and Space Administration (NASA). KORUS-

AQ included twenty flights using the NASA DC-8 aircraft from May 1 to June 9, 2016, complemented by heavily instrumented ground sites including aerosol composition at Olympic Park and the Korea Institute of Science and Technology (KIST) in Seoul. The NIER maintains the extensive AirKorea monitoring network for hourly observations of $PM_{2.5}$ mass, ozone, and other pollutants, with 329 sites available during the campaign, including locations near Olympic Park and KIST. There were four distinct meteorological periods during KORUS-AQ, described in Peterson et al. (2019). These included a dynamic period

characterized by a series of frontal passages (Dynamic Period, May 1-16), dry, clear, and stagnant conditions (Stagnant Period, May 17-22), long-range transport and haze conditions with high humidity and extensive cloud cover (Transport/Haze Period, May 25-31), and blocking conditions limiting transport (Blocking Period, June 1-7). Details on the impact of the different meteorological periods on $PM_{2.5}$ are provided in Jordan et al. (2020). We focus on the Seoul Metropolitan Area (SMA) with the highest density of KORUS-AQ observations and the highest $PM_{2.5}$ levels observed by the AirKorea network during the

campaign. Crawford et al. (2021) provides a full listing of all observations made during KORUS-AQ. Table 1 describes the aircraft and ground observations used in this work.



# 3 GEOS-Chem model

We use the GEOS-Chem chemical transport model (CTM) in version 12.7.2 (doi: 10.5281/zenodo.3701669) to simulate KORUS-AQ. The model is driven by assimilated meteorological data from the NASA Global Modeling and Assimilation Office (GMAO) Goddard Earth Observing System Forward-Processing (GEOS-FP) atmospheric data assimilation system. GEOS-FP has a native horizontal resolution of $0.25° \times 0.3125°$, which we apply with the nested version of GEOS-Chem (Chen et al., 2009) over East Asia (70° - 140°E, 15°S - 55°N) using boundary conditions from a global simulation at $2.0° \times 2.5°$ with a 1-month initialization period. The model has 47 vertical layers, with the first layer centered at approximately 60 m above the surface. Model timesteps are 20 min (chemistry) and 10 min (transport) as recommended by Philip et al. (2016).

Global emissions are from the Community Emissions Database System (CEDS) inventory (Hoesly et al., 2018) overwritten by the KORUSv5 anthropogenic and shipping emissions (Woo et al., 2020) for Asia (60° - 146°E, 10°S - 54°N) developed for the KORUS-AQ campaign. The translation from KORUSv5, provided using the SAPRC99 chemical mechanism, to the GEOS-Chem mechanism is given in Table S1. We apply sector-specific diurnal variation from the Multi-resolution Emission Inventory for China (MEIC) as in Miao et al. (2020) to the monthly KORUSv5 emissions. Natural emissions are from the Global Emissions Initiative (GEIA, Bouwman et al., 1997) for ammonia and from MEGANv2.1 (Guenther et al., 2012) for biogenic species. We include lightning emissions (Murray et al., 2012), biomass burning emissions (GFED4s, Werf et al., 2017), soil $NO_x$ emissions (Hudman et al., 2012), and volcanic $SO_2$ emissions (Carn et al., 2015). Table 2 shows the emissions inventory for key emitted species in the nested East Asia domain for May 2016.

Model dry deposition for gas-phase species is based on the resistance-in-series scheme from Wesely (1989) as implemented by Wang et al. (1998), where species deposition is limited by aerodynamic resistance, quasi-laminar layer resistance, and canopy or surface resistance. Species with low surface resistance, such as $HNO_3$, are limited in their deposition velocity by aerodynamic resistance only. Aerosol deposition is from Zhang et al. (2001). The original model wet deposition scheme is described by Liu et al. (2001) for water-soluble aerosols and Amos et al. (2012) for gases. Wet deposition includes scavenging from moist convective updrafts and rainout and washout from precipitation. We include the revised wet deposition scheme of Luo et al. (2019) that uses an empirical washout rate for nitric acid two orders of magnitude higher than the previous value and replaces the standard constant value for in-cloud condensation water content with the value calculated by the meteorological fields (GEOS-FP). GEOS-Chem uses a non-local boundary layer mixing scheme (Holtslag and Boville, 1993; Lin and McElroy, 2010) where mixing is calculated explicitly from meteorological variables provided by GEOS-FP (i.e. sensible and latent heat flux, temperature, friction velocity). The mixing height is restricted from dropping below a minimum mechanical mixing depth, defined as a function of local friction velocity (Lin and McElroy, 2010).





The GEOS-Chem HOₓ-NOₓ-VOC-ozone-halogen-aerosol mechanism includes improvements to PAN chemistry (Fischer et al., 2014), isoprene oxidation (Fisher et al., 2016; Travis et al., 2016; Chan Miller et al., 2017), halogen chemistry (Sherwen et al., 2016), Criegee intermediates (Millet et al., 2015), and methyl, ethyl, and propyl nitrates (Fisher et al., 2018). Heterogeneous aerosol uptake of $HO_2$ produces $H_2O_2$ (Mao et al., 2013), with a reactive uptake coefficient ($\gamma$) of 0.2 (Jacob, 2000). We implement aromatic chemistry from Yan et al. (2019) for the simulation of KORUS-AQ.

We use the model "simple scheme" for organic aerosol (OA) where OA is generated using fixed empirically derived yields from isoprene, monoterpenes, biomass burning, and anthropogenic fuel combustion (Pai et al., 2020). This scheme includes an emitted hydrophobic component (OCPO) with an assumed organic-mass-to-organic carbon (OM:OC) ratio of 1.4 that is aged to a hydrophilic oxygenated component (OCPI) with an OM:OC ratio of 2.1. Secondary organic aerosol (SOA) is a lumped product (SOAS) with a molecular weight of 150 g mol⁻¹. For comparison to observations, primary organic aerosol (POA) is defined as OCPO and SOA is the sum of OCPI and SOAS. The sulfate-nitrate-ammonium (SNA) aerosol simulation (Park, 2004) includes the addition of metal-catalyzed oxidation of $SO_2$ (Alexander et al., 2009), sulfur oxidation by reactive halogens (Chen et al., 2017), and improved implementation of aerosol cloud-processing and revised uptake coefficients for $NO_2$ (Holmes et al., 2019). Uptake of $N_2O_5$ on SNA includes dependence on aerosol water, organic coatings, nitrate aerosol fraction, and particulate chloride (McDuffie et al., 2018). SNA partitioning is calculated with ISORROPIA v2.2 (Pye et al., 2009). The model includes accumulation mode (SALA) and coarse mode (SALC) sea salt aerosol (Alexander et al., 2005; Jaeglé et al., 2011) and dust in four size bins (DST1 to 4) (Fairlie et al., 2010), where the first bin and 38% of the second bin are included in $PM_{2.5}$. The recommended definition of dry $PM_{2.5}$ is given by Eq 1.

$$PM_{2.5} = SO_4^{2-} + NO_3^- + NH_4^+ + BC + OCPO * 1.4 + OCPI * 2.1 + SOAS + SALA + DST1 + DST2 * 0.38, \qquad (1)$$

The AirKorea $PM_{2.5}$ observations provided by NIER are obtained using the beta-ray attenuation method (BAM-1020, Table 1). We do not adjust modeled $PM_{2.5}$ for any measurement relative humidity effects as the BAM-1020 has been shown to perform well against federal reference method monitors (Le et al., 2020).

Specific details of production of model nitric acid ($HNO_3$), the gas-phase precursor to aerosol nitrate ($NO_3^-$ = pNO3), are provided below as KORUS-AQ provides detailed observations of this chemistry. Reactions R1-R6 describe model production of $HNO_3$ from oxidation of $NO_2$ (R1), aqueous uptake and reaction of $N_2O_5$, $NO_2$, and $NO_3$ on aerosol (R2, R4, R5), aqueous uptake and reaction of $N_2O_5$ and $NO_3$ in cloud water (R3, R5), heterogeneous halogen chemistry (Table S2), and oxidation of VOCs by the nitrate radical (R6). In R2, aqueous uptake and reaction of $N_2O_5$ with particle chloride (Cl⁻) produces nitryl chloride ($ClNO_2$) with a yield ($\emptyset$) of 1 on sea salt aerosol and zero on all other aerosol types.

$$NO_2 + OH \xrightarrow{M} HNO_3 \qquad (R1)$$



$$N_2O_5 \xrightarrow{aerosol} (2 - \emptyset)HNO_3 + \emptyset ClNO_2 \qquad (R2)$$

$$N_2O_5 \xrightarrow{cloud} 2HNO_3 \qquad (R3)$$

$$NO_2 \xrightarrow{aerosol} 0.5HNO_3 + 0.5HNO_2 \qquad (R4)$$

$$NO_3 \xrightarrow{aerosol/cloud} HNO_3 \qquad (R5)$$

$$NO_3 + VOC \rightarrow HNO_3 \qquad (R6)$$

## 4 Simulation of PM$_{2.5}$ during KORUS-AQ

Figure 1a shows the model simulation of daily average PM$_{2.5}$ (Eq. 1) compared to the observed average of the 15 AirKorea sites within the GEOS-Chem grid box containing the major SMA monitoring sites (KIST and Olympic Park). These two sites are in close proximity to the AirKorea monitors (Fig. 1b). Campaign average PM$_{2.5}$ is 29 µg m$^{-3}$, but this increases to 53 µg m$^{-3}$ during the Transport/Haze period. The model reproduces the low PM$_{2.5}$ during the Dynamic period, the increase during the Transport/Haze period, and the variable concentrations during the Blocking period. Across the campaign, the model underestimates PM$_{2.5}$ (NMB = -15%) due to a low bias during the Stagnant period and the initial build-up during the Transport/Haze period. This model performance is similar to Choi et al. (2019) using a different GEOS-Chem configuration.

Figure 2 compares observed PM$_{2.5}$ composition against the model for the gridbox containing the KIST ground site. Measured composition from the KIST HR-ToF-AMS instrument (Table 1), representative of PM$_1$ (Guo et al., 2021), is used to speciate daily average PM$_{2.5}$ from the AirKorea sites (Fig. 1a). Jordan et al. (2020) showed that speciated PM$_1$ was generally representative of PM$_{2.5}$ mass throughout KORUS-AQ, except for the Transport/Haze period when PM$_{2.5}$ significantly exceeded PM$_1$. The strong correlation between PM$_{2.5}$ and PM$_1$ during the campaign implied growth of PM$_1$ to larger sizes. Dust is not a major component of PM$_{2.5}$ at the surface after May 9$^{th}$, as further discussed in Section S1. Therefore PM$_1$ composition likely represents the composition of PM$_{2.5}$ with the exception of a small contribution from primary aerosol species. Sun et al. (2020) showed that PM$_{2.5}$ can be up to 50% greater than PM$_1$ in polluted, humid environments and the mass at sizes >PM$_1$ is secondary (not BC or POA). We remove BC and POA from observed PM$_{2.5}$ and scale the remaining components (SNA, SOA) to the remaining PM$_{2.5}$. The resulting speciated PM$_{2.5}$, derived from KIST PM$_1$ composition and AirKorea PM$_{2.5}$ mass, is provided for each meteorological period in Table 3. Figure 2 and Table 3 include the ALWC associated with PM$_{2.5}$, calculated for the observations using the E-AIM IV thermodynamic model (Clegg and Brimblecombe, 1990; Clegg et al., 1998; Massucci et al., 1999; Wexler and Clegg, 2002), and ISORROPIAv2.2 (Pye et al., 2009) in GEOS-Chem. During KORUS-AQ, Kim et al. (2022) found that ISORROPIAv2.2 provided similar results as the E-AIM model, reproducing E-AIM pH within ~0.4 units.

The primary campaign average model biases are underestimated sulfate (-64%), overestimated nitrate (+36%), and underestimated SOA (-43%). The excess model nitrate is the primary driver of overestimated ALWC (+82%). During the



Stagnant period, the model low bias is due to underestimated SOA (-9 µg m$^{-3}$). This may be due to missing local production from emissions of semi- and intermediate-volatility volatile organic compounds (S/IVOCs, McDonald et al., 2018) and aromatics (Nault et al., 2018), primarily attributable to solvents and vehicle emissions (Shin et al., 2013a, 2013b; Simpson et al., 2020). During the Dynamic and Blocking periods, the model PM$_{2.5}$ bias is within 20% of the observations but with overestimated nitrate and underestimated sulfate. The model severely underestimates sulfate during the Transport/Haze period (-11 µg m$^{-3}$, Table 3) suggesting that the model fails to reproduce the processes driving the pollution episode. As described by Jordan et al. (2020), both ground and aircraft observations during KORUS-AQ showed that cloudy and humid conditions during the Transport/Haze period increased PM$_{2.5}$ through heterogeneous production of SNA.

The KORUS-AQ aircraft observations included detailed daytime (available from ~8am to 4pm KST) aerosol and gas-phase observations that we use to determine the cause of model sulfate and nitrate biases and their regional extent. Model SOA biases will be the subject of future work as here they do not contribute to PM$_{2.5}$ exceedances (50 µg m$^{-3}$ daily average in 2016). The KORUS-AQ campaign included frequent sampling along a repeated flight pattern or "stereoroute" over the SMA up to three times a day, supplemented by less frequent flights to investigate specific source regions or transport events (Crawford et al., 2021). Figure S2 shows the high data density in the SMA compared to the rest of the study region. We use the 55 descents over Olympic Park from the SMA stereoroute to compare against the daily surface observations shown in Fig. 2.

Figure 3 shows the mean daytime aircraft profiles of sulfate and nitrate for the descents over Olympic Park below 2 km separated by the same meteorological periods as Fig. 2. The corresponding profiles for SO$_2$ and nitric acid are shown in Fig. S3. The model is sampled along the flight tracks and both the model and the observations are averaged to the model grid, timestep and nearest vertical 0.5 km. Similar to the daily surface average, the model underestimates daytime sulfate below 2 km with the most severe bias (-8 µg m$^{-3}$ in the lowest altitude bin of 0.5 km) occurring during the Transport/Haze period. Unlike in the daily surface average, the model underestimates daytime nitrate below ~1 km with the exception of the Dynamic period when nitrate is in good agreement. The model nitrate underestimate could be partially related to the low bias in model RH of up to -8% (Blocking period, 39 vs. 47%) below 0.5 km (Fig. S3) or overestimated mixed layer height (Oak et al., 2019). If the model RH simulation was unbiased, we would expect an improved simulation of nitrate as the minimal RH bias during the Dynamic period corresponds to the best nitrate simulation (Fig. 3, Fig. S3). Model aerosol dry deposition may also be too fast but this effect would increase model concentrations by only ~10% (Emerson et al., 2020).

There is no available measurement of PM$_{2.5}$ from the aircraft to provide a similar scaling from PM$_1$ to PM$_{2.5}$ as was done in Fig. 2. However, any increase to the observed profiles of PM$_1$ sulfate or nitrate to account for possible growth to larger sizes would exacerbate the model underestimate of these species. The discrepancy between the model low to minimal bias against daytime aircraft nitrate observations (Fig. 3) and the overestimate against daily average nitrate at the KIST ground site (Fig.



2) implies a failure of the model to represent nighttime chemical production. We investigate the possible causes of overestimated daily average model nitrate in Section 5 and underestimated model sulfate in Section 6.

## 5 Model errors representing the nitrate diurnal cycle

The discrepancy between the model daytime and daily performance for nitrate demonstrates the need to compare the model mean nitrate diurnal cycle against the nitrate fraction of $PM_{2.5}$ derived from KIST observations as described in Section 4. Figure 4a shows that between 6am and 6pm KST (daytime) the model bias minimal ($< -1$ µg m$^{-3}$) while the bias from 6pm to 6am KST (nighttime) is $+3$ µg m$^{-3}$. As described in Section 3, the model has a newly revised treatment of wet scavenging that significantly reduces the model nitrate and nitric acid biases present in previous model versions (Luo et al., 2019). Without this improvement, the model would have an average nighttime bias of $+7$ µg m$^{-3}$. Figure S4 shows daily precipitation in Seoul from the Korea Meteorological Administration (KMA, 2021) which is infrequent and negligible in the later part of the campaign. The model underestimate in total precipitation across the campaign is minimal (121 vs. 112 mm). Insufficient wet scavenging is unlikely to be the cause of the remaining model nitrate bias.

We perform a sensitivity test to determine the relative impact of daytime (R1) vs. nighttime (R2-R5, Section 3) production of $HNO_3$ on the model bias by shutting off the nighttime reactions. Figure 4c shows that the main model nighttime pathway is aerosol uptake of $NO_2$ (R4) with a small contribution from $N_2O_5$ hydrolysis (R2/3) in the early morning hours. Figure 4a shows that removing nighttime chemistry results in improved early morning agreement (1am to 8am KST) but the evening overestimate (8pm to 1am KST) is less affected. Jordan et al. (2020) showed observational evidence for significant nighttime production of nitrate by $N_2O_5$ hydrolysis (R2). We use the removal of nighttime chemistry to hypothesize that part of the model nighttime bias is due to excess daytime $HNO_3$ that has not yet been lost to deposition and is converted to nitrate as conditions become thermodynamically favorable for partitioning to the aerosol-phase. The dominance of $NO_2$ uptake over $N_2O_5$ hydrolysis in the model suggests that there are additional errors in simulated nighttime chemistry.

### 5.1 Sensitivity of model nitrate bias to gas-phase precursors

Inorganic aerosol ammonium nitrate ($NH_4NO_3$) is formed by dissolution of $HNO_3$, which reacts in the aqueous phase with ammonia ($NH_3$) to establish a thermal equilibrium with $NH_4NO_3$. The conditions that favor $NH_4NO_3$ are generally cool and humid (i.e., nighttime) and characterized by high $NH_3$ and $HNO_3$ concentrations relative to sulfate (Guo et al., 2016). We calculate that average nighttime RH (temperature) in the SMA is 74% (290K) compared to the model value of 71% (288K), indicating that significant errors in RH or temperature are not the cause of nighttime biases. Overproduction of model nighttime nitrate could be due to overestimated $NH_3$ if this species limits $NH_4NO_3$ production. In South Korea, and generally East Asia, $NH_4NO_3$ is limited by availability of $HNO_3$. This due to high levels of $NH_3$ (~10 ppb) observed in East Asia, attributable to non-agricultural sources such as transportation (Song et al., 2009; Phan et al., 2013; Link et al., 2017; Sun et al., 2017; Chang



et al., 2019). The model reproduces the expected high concentration of $NH_3$ with an average of 9 ppb at Olympic Park. Ibikunle et al. (2020) performed a rigorous thermodynamic assessment of KORUS-AQ observations confirming that aerosol was always sensitive to $HNO_3$ in polluted conditions. Nitrate-limited SNA thermodynamics were observed in similar conditions in China and successfully represented by ISORROPIA v2.2 in GEOS-Chem (Zhai et al., 2021).

Few datasets exist to further test the performance of $HNO_3$-$pNO_3$ partitioning in the model but KORUS-AQ observations provide this opportunity. This partitioning is described by Eq. 2, where the ratio of $pNO_3$ to total nitrate ($TNO_3 = HNO_3$ and $pNO_3$), known as $\varepsilon NO_3$, is impacted by temperature, relative humidity, and aerosol composition (Guo et al., 2016, 2017).

$$\varepsilon NO_3 = \frac{pNO_3}{HNO_3 + pNO_3} \tag{2}$$

Accurate simulation of $\varepsilon NO_3$ is critical to regulating the deposition of $TNO_3$ as $HNO_3$ deposits more rapidly than $pNO_3$ (Nenes et al., 2021). Figure 5 shows $\varepsilon NO_3$ as a function of RH for the observations and the model for the same domain as Fig. 3 below 1.5 km. While the model represents the increase of $\varepsilon NO_3$ with RH, model $\varepsilon NO_3$ is generally underestimated, particularly at lower RH (<50%). This low bias in $\varepsilon NO_3$ could be due to overestimated $HNO_3$, as the lower RH and associated higher temperatures generally prevent excess $HNO_3$ (denominator of Eq. 2) from partitioning to the aerosol-phase. We discuss the possibility of overestimated model $HNO_3$ below. As $\varepsilon NO_3$ is underestimated in the model, excess partitioning to the aerosol-phase is not a cause of the model nitrate overestimate shown in Fig. 2. The successful performance of ISORROPIAv2.2 during KORUS-AQ is also evident from the comparison against the E-AIM model in Kim et al. (2022).

Figure 6a shows vertical profiles of observed and modeled $HNO_3$ for the Olympic Park descents. The model overestimates $HNO_3$ in the lowest bin (0.5 km) by +1600 ppt or +100%. This high bias persists across most of the study domain except over the ocean south of 34°N (Fig. 7) where local emissions have a small impact and loss to deposition is slow. During average daytime conditions (~50% RH, 295K), model $\varepsilon NO_3$ is ~0.3, indicating that while the aerosol is $HNO_3$-limited, higher temperatures and low RH also prevent the excess model $HNO_3$ from partitioning to aerosol. A simulation turning off South Korean emissions shows that local sources contribute ~50% to model $HNO_3$ concentrations below 0.5 km (Fig. 6a). Thus while model errors in emissions or chemistry could be a cause of the bias, an overestimated lifetime of $HNO_3$ against dry or wet deposition could also play a role. We evaluate these possibilities further in Section 5.2.

**5.2 Causes of overestimated daytime $HNO_3$**

KORUS-AQ provides aircraft and surface observations that provide additional constraints on the model $HNO_3$ bias of +100% described in Section 5.1. We use observations of $NO_2$ and OH from aircraft to evaluate whether $NO_x$ emissions or production from R1 ($NO_2$ + OH) are overestimated. Figure 6b shows that model $NO_2$ is underestimated by -40% below 0.5 km. This is partially due to the expected model inability to resolve the highest observed levels of $NO_2$ in an urban region, illustrated by the larger standard deviation in the observations compared to the model. However, given the same emissions inventory used



here (KORUSv5), a set of eight models varied in their biases for $NO_x$ against KORUS-AQ aircraft observations from a minimal underestimate (-7%) to a large overestimate (+56%) depending on model configuration (Park et al., 2021). Thus, model biases could be due to a range of factors including underestimated emissions, inaccuracies in the emission diurnal cycle, or overestimated mixed layer heights. Errors in any of these factors that could increase model $NO_2$, such as decreased mixed layer heights or increased emissions, would be expected to increase the model overestimate of $HNO_3$. Fig 6c shows that the model bias in OH is small (+20%) and well within measurement uncertainty (+32%) and therefore it is unlikely that model errors in R1 could cause the model $HNO_3$ bias +100%.

The fastest removal pathways for $HNO_3$ are wet and dry deposition. The model implementation of these processes is described in Section 2. The revised wet scavenging scheme has improved annual average model simulations of $HNO_3$, but the effect on $HNO_3$ during KORUS-AQ is limited as precipitation was infrequent after the beginning of the campaign as discussed above. Section S2 further discusses the impact of this scheme on KORUS-AQ nitrate and $HNO_3$ but errors in wet deposition are unlikely to be the cause of overestimated model $HNO_3$. Section S2 also describes other possible loss pathways to dust, seasalt, or production of $ClNO_2$ from $N_2O_5$ hydrolysis that have negligible effects on the model $HNO_3$ and nitrate.

Previous attempts to improve model nitrate invoked an unknown sink of $HNO_3$ in the model (Heald et al., 2012; Weagle et al., 2018), as uncertainties in precursor emissions, the rate of $N_2O_5$ hydrolysis (R2/R3) or gas-phase production (R1), OH concentrations, and $HNO_3$ dry deposition velocity ($Vd_{HNO3}$) could not explain model nitrate biases. We similarly conclude that an unknown loss process must be a main cause of the daytime model overestimate in $HNO_3$ and associated evening nitrate bias during KORUS-AQ that occurs as conditions become more favorable for partitioning $HNO_3$ to $pNO_3$. This unknown loss process could be a larger underestimate in dry deposition than has been previously considered, as constraints from KORUS-AQ show that uncertainties in emissions, nighttime production (R1-R5), and wet deposition are not the cause. Heald et al. (2012) ruled out dry deposition after assuming an uncertainty of a factor of two. Here, the increase in $Vd_{HNO3}$ required to reproduce observed $HNO_3$ (Fig. 6a) is a factor of five. A similar increase in $Vd_{HNO3}$ was invoked by Itahashi et al. (2017) in their model study of wintertime nitrate in East Asia based on the finding from Shimadera et al. (2014) that $Vd_{HNO3}$ (as well as $NH_3$ emissions and dry deposition) were the main factors driving model nitrate performance.

The increase in $Vd_{HNO3}$ suggested above would result in an average value of 7.5 cm $s^{-1}$ compared to the standard model value of 1.5 cm $s^{-1}$. This corresponds to a maximum midday rate of 15.4 cm $s^{-1}$ cm $s^{-1}$ compared to the original value of 3.1 cm $s^{-1}$ (Fig S8). Deposition of $HNO_3$ is limited only by aerodynamic resistance (and available surface area), as it readily adheres to surfaces. While the increase to $Vd_{HNO3}$ we suggest here is large, this could arise from factors such as increased surface area in urban or heavily forested regions and increased vertical mixing over cities due to turbulence induced by the urban heat island effect. These factors are not accounted for in the limited existing deposition velocity measurements that have been compared against models (Nguyen et al., 2015). Increased turbulence over forested regions results in higher deposition velocities





(Sievering et al., 2001; Yazbeck et al., 2021), which would also be expected in an urban environment (i.e. Keuken et al., 1990). The model does not account for increased available surface area for deposition contributed by urban buildings, or the elevated vertical mixing over cities due to the urban heat island effect (Hong and Hong, 2016; Halios and Barlow, 2018). Dry deposition rates thus may be much higher than in model parameterizations that do not include a specific treatment of the urban canopy (Cherin et al., 2015) and this is the case in GEOS-Chem.

Neuman et al. (2004) derived $Vd_{HNO3}$ from aircraft observations of power plant plumes in eastern Texas, obtaining values between 8 and 26 cm s$^{-1}$, values at least four times faster than reported previously. We take a similar approach to Neuman et al. (2004) to calculate $Vd_{HNO3}$ from KORUS-AQ observations in the SMA using the rate equation for TNO$_3$ as a function of photochemical age (Fig. 8, Eq. 3).

$$TNO_3(t) = \frac{NO_x(0)}{\frac{\beta}{c}-1}\left(e^{-ct} - e^{-\beta t}\right) \tag{3}$$

$NO_x(0)$ is the initial NO$_x$ mixing ratio normalized to CO (Fig. 8, 0.24 ppbv / ppbv CO), $\beta$ is the first order loss rate for TNO$_3$, c is the first order production rate for TNO$_3$ (pTNO$_3$ = pHNO$_3$ = $k_{R1}$[OH]), and TNO$_3$(t) is observed TNO$_3$ as a function of photochemical age (t). As the production of TNO$_3$ was constrained by observed OH, and assuming the main loss of TNO$_3$ ($\beta$) is from deposition of HNO$_3$, the unknown for TNO$_3$ evolution is the deposition rate. The full details of this calculation are provided in Section S3.

Figure 8 shows NO$_x$, TNO$_3$, and the other NO$_x$ oxidation products of total peroxy nitrates ($\Sigma$PNs) and the sum of alkyl- and multi-functional nitrates ($\Sigma$ANs) as a function of photochemical age. All species are normalized by background subtracted CO. NO$_x$ is continuously depleted at a rate of 0.31 hr$^{-1}$, implying continued production of TNO$_3$, $\Sigma$PNs, and $\Sigma$ANs. This loss rate corresponds to a lifetime of 3.2 hrs that is similar to the lifetime of 4.8 hrs for NO$_2$ against conversion to HNO$_3$ (R1) using the SMA average OH of $5.2\times10^{-6}$ molec cm$^{-3}$. From Eq. 3, we derive a loss rate ($\beta$) of 13.9 cm s$^{-1}$ that best fits the observed change in TNO$_3$ with aging. As deposition of pNO$_3$ is slow, we assume that $Vd_{HNO3}=Vd_{TNO3}$. All three NO$_x$ oxidation products (TNO$_3$, $\Sigma$PNs, $\Sigma$ANs) exhibit similar behavior with production outpacing loss until approximately three hours of aging, where loss appears to balance production and concentrations remain relatively constant. There is likely large uncertainty in the derived photochemical ages shown in Fig 8, as the aircraft did not follow plumes as in Neuman et al. (2004). However, our derived NO$_x$ lifetime is consistent with average SMA conditions and is not affected by our choice of observed altitude range, suggesting that the aging represents true chemical processing.

Figure 8 shows that the slower value for midday $Vd_{HNO3}$ in the original model (3.1 cm s$^{-1}$) poorly represents observations compared to the faster value obtained in Fig. 6 (15.4 cm s$^{-1}$). We calculate that the original deposition rate would correspond to a first order loss rate for TNO$_3$ of only 0.07 hr$^{-1}$ (assuming a 1.5 km boundary layer height) and thus observed TNO$_3$ should increase with photochemical age, which is not supported by the observed relationship in Fig. 8. The factor of five increase in


Vd$_{HNO3}$, constrained only using observed HNO$_3$, implies a similar loss rate of TNO$_3$ as derived in Fig. 8 and leads to the observed behavior where after initial production, the normalized mixing ratio remains constant. This analysis supports the hypothesis given above, that existing observations supporting lower values for Vd$_{HNO3}$ (Nguyen et al., 2015) may underrepresent deposition in regions with greater turbulence and available surface area such as in cities like Seoul. Deposition of atmospheric pollutants such as nitric acid on buildings generates 'urban grime' that may photolyze and produce NO$_x$ and

HONO (Baergen and Donaldson, 2013, 2016). This urban grime could be a source of HONO (Zhang et al., 2016) and may be larger than previously thought if models underestimate nitric acid deposition.

Figure 4a shows the impact to the diurnal cycle of model nitrate from increasing model Vd$_{HNO3}$ by a factor of five. The rapid late afternoon /early evening increase in model nitrate (Fig. 4a) is largely resolved and the model simulation of HNO$_3$ is now

in good agreement with aircraft observations (Fig. 6) due to a significant dampening of the HNO$_3$ diurnal cycle (Fig. 4b). This reduction in the HNO$_3$ diurnal cycle is better supported by observations of TNO$_3$ as discussed above. We conclude that a key reason for the overestimated daily average model nitrate shown in Fig. 2 is overestimated daytime HNO$_3$ that results in excess nighttime nitrate when conditions become favorable (cool and humid) for gas to aerosol partitioning. The model overestimate is due to insufficient loss, likely underestimated dry deposition. This finding does not address possible errors in model

00   nighttime production pathways (NO$_2$ vs. N$_2$O$_5$), and KORUS-AQ provides detailed ground observations that can be used to constrain the model representation of nighttime chemistry.

## 5.3 Errors in model nighttime production of HNO$_3$

Figure 4c shows that model nighttime production of HNO$_3$ by aerosol uptake of NO$_2$ (R4) is approximately twice as large as R2 (N$_2$O$_5$ hydrolysis). This contradicts the calculation from Jordan et al. (2020) that R2 is the driver of nitrate production

05   during KORUS-AQ, particularly during the Transport/Haze period due to sufficient nighttime ozone concentrations that allow for production of the nitrate radical and N$_2$O$_5$ through R8 and R9.

$$O_3 + NO \rightarrow NO_2 \tag{R7}$$
$$NO_2 + O_3 \rightarrow NO_3 + O_2 \tag{R8}$$
$$NO_2 + NO_3 \xrightarrow{M} N_2O_5 \tag{R9}$$

10   Production of nitrate by N$_2$O$_5$ hydrolysis is supported by observations of ClNO$_2$, thought to be produced primarily by this reaction (Thornton et al., 2010). As discussed above in Section 5.2, observations of ClNO$_2$ at Olympic Park are elevated at night (Fig. S7). Despite recent large reductions of the uptake coefficient ($\gamma$) for NO$_2$ in the model (Holmes et al., 2019), NO2 uptake still is the dominant nighttime pathway for HNO$_3$ production in the model. We use observations of ozone, NO, and NO$_2$ at Olympic Park to determine whether errors in R7-R9 are impacting model ability to produce N$_2$O$_5$.



Figure 9 shows the mean modeled and observed diurnal cycles of ozone and $NO_2$ for the AirKorea sites in the model grid box (Fig. 1b) and for ozone, NO, $NO_2$, and $NO_x$ at Olympic Park. Ozone might be expected to be titrated in an urban area by R7 as the mixed layer collapses in the evening, resulting in elevated NO and shutting down production of the nitrate radical (R8). This is the case in the model where nighttime ozone is <2 ppb approximately 20% of the time but this never occurs in the observations (Fig. S9). As a result, average observed nighttime ozone is 24 ppb but only 13 ppb in the model (Fig. 9). The time series of observed and modeled ozone in Fig. S9 shows while the model does succeed in simulating high nighttime ozone concentrations during the Dynamic Period, characterized by higher windspeeds, ozone is incorrectly titrated at other times particularly during the buildup of the haze pollution following a frontal passage on May 24th. The implications of this excess ozone titration for the simulation of $PM_{2.5}$ specifically during haze conditions will be further discussed in Section 6.

As shown in Fig. 9b+c, model ozone titration corresponds to excess model NO and $NO_2$ at night and explains the dominance of $NO_2$ uptake in the model over $N_2O_5$ hydrolysis for nighttime $HNO_3$ production. The model bias for $NO_x$ is minimal during the day, providing additional support for the level of emissions in the model, but is overestimated by a factor of two at night. The excess model ozone titration and overestimated nighttime $NO_x$ implies an error in nocturnal mixing. Figure 10a shows the mixed layer height (MLH) diurnal cycle measured by ceilometers at Olympic Park and Seoul National University. The aerosol gradients detected by the ceilometer to estimate MLH are less reliable at night due to the possible presence of aerosols in the residual layer (Jordan et al., 2020). We support these measurements with additional calculations of nighttime MLH from radiosonde observations of temperature and RH four times a day (Section S4, Fig. S10), showing that the average MLH at 3 KST could be ~300m compared to 220m in the model. As previously discussed in Section 5.2, in urban regions such as Seoul, the anthropogenic heat island effect and the heterogeneity of the urban land cover increase sensible heat fluxes and turbulence over non-urban areas (Halios and Barlow, 2018) and create an unstable mixed layer even at night. Min et al. (2020) showed that the nighttime mixed layer in Seoul is elevated in all seasons, and that nighttime conditions are generally unstable due to urban heat storage and anthropogenic heat release and this could explain the observed elevated nighttime MLH (Fig. 10a, Fig. S11). This effect is not captured in many meteorological models including the one used here (GEOS-CF, Section 3). Nighttime sensible heat flux in the model is always negative (stable conditions) (Fig. 10b).

Starting at 17 KST, the model mixed layer collapses early, causing a more rapid decline in ozone than in the observations (Fig. 9a, Fig. 10a). The transition from convective daytime mixed-layer to stable nocturnal boundary layer is poorly understood (Lothon et al., 2014). The early collapse of the mixed layer has been observed in other models including the widely-used Weather Research and Forecasting (WRF) model over the Baltimore-Washington, D.C. region during the NASA DISCOVER-AQ mission (Hegarty et al., 2018). One possibility for the delay in this collapse is continued mixing from the last eddy of the day formed just before the sensible heat flux changes sign during the evening transition (Blay-Carreras et al., 2014). This has been hypothesized as reasons for errors in the model diurnal cycle of ozone in the Southeast United States (Travis and Jacob, 2019). Here, this early collapse drives excess production of nitrate from $NO_2$ (R4).



·50

While addressing the shortcomings of the model mixing scheme is beyond the scope of this study, we test the sensitivity of model nitrate production to the main two problems identified above, 1) the overly rapid collapse of the afternoon mixed layer, and 2) insufficient nocturnal mixing. While model meteorology is calculated offline, mixing in the boundary layer is calculated online (Section 2), allowing us to perturb mixing parameters. We increase the nighttime MLH to 500m to examine the impact

·55     on model ozone, NO, and $NO_2$. The effect of this change on these species is minimal (Fig. 9), similar to the findings of other model sensitivity studies that performed this same test (Oak et al., 2019; Miao et al., 2020). While the strength of model vertical mixing is sensitive to MLH, the model sensible heat flux and friction velocity have a larger impact (Holtslag and Boville, 1993), and the nighttime mixed layer will remain stable while the sensible heat flux is negative regardless of MLH.

·60     The increase in nighttime mixing in urban vs. rural regions has been addressed in the CMAQ model (Li and Rappenglueck, 2018) by using a higher value for the minimum mixing strength (eddy diffusivity) over urban areas. However, we find that this approach is insufficient to address model ozone titration without increasing the sensible heat flux to a positive value to produce an unstable mixed layer. This is illustrated in Fig. S12, where we scale the model MLH to match the profile at Olympic Park (Fig. 10a) and raise the model minimum eddy diffusivity from 0.01 $m^2$ $s^{-1}$ to 1 $m^2$ $s^{-1}$ over the SMA. Reducing the collapse

·65     of the evening MLH without a change to the drivers of mixing (i.e., heat fluxes, friction velocity) has negligible impact on decreasing model ozone titration (Fig. S12). In addition, the MLH at Olympic Park in the early morning hours appears inconsistent with observed ozone, likely due to the uncertainties in the measurement technique discussed above and supported by the lower values obtained from radiosonde profiles (Fig. S11). Errors in model nighttime mixing are difficult to remedy without significant revisions to the model mixing parameterizations, including implementing continued mixing from daytime

·70     eddies into the evening hours (Blay-Carreras et al., 2014) and parameterizing the excess sensible heat flux in urban areas (Halios and Barlow, 2018). We address the implications of these errors in the simulation of haze pollution events in Section 6.

## 6 Model simulation of haze buildup

The failure of models to simulate sulfate production in haze in East Asia is a current topic of intensive research and is attributable to missing sulfate production in aerosol water (Wang et al., 2014; Zheng et al., 2015a; Chen et al., 2016; Shao et

·75     al., 2019; Miao et al., 2020). There has been less attention paid to the ability of models to simulate nitrate in haze as nitrate-dominated haze is a more recent phenomenon due to the reductions in $SO_2$ in East Asia (Wang et al., 2020). Figure 2 and Table 3 show that the model can reproduce the increase in the nitrate component of $PM_{2.5}$ during the Transport/Haze period but overestimates absolute concentrations by ~20%. This contributes to an 80% overestimate in ALWC. Efforts to explicitly simulate $SO_2$ oxidation in ALWC may be hindered by this model bias, which also impacts the rates of all other heterogeneous

·80     reactions through the increase in aerosol surface area.



Figure 11a shows the hourly time series of observed and modeled nitrate at Olympic Park during the Transport/Haze period. During the haze buildup, the model initially overestimates nitrate during the day (5/24) followed by large nighttime underestimates (5/24-5/25). This is opposite to the nighttime overestimate but daytime agreement shown in the campaign average (Fig. 4a). During the haze buildup, daytime RH remained elevated (>50%, Fig. S13) and the daytime mixed layer was suppressed (Fig. S14 and Jordan et al. 2020). The model reproduces both conditions, which are favorable for SNA production. Model nitrate biases here are likely due to the errors identified in Section 5.2 (overestimated daytime $HNO_3$) and Section 5.3 (incorrect representation of nighttime conditions), but here the excess daytime $HNO_3$ in the model results in higher daytime nitrate than in the campaign average. Insufficient model sulfate during the haze event results in overestimated model pH and excess partitioning of $HNO_3$ to the particle phase (Guo et al., 2016). Fig. S15 shows that $\epsilon NO_3$ (the calculated fraction of $TNO_3$ in the aerosol phase) decreases as sulfate increases and the model sulfate bias corresponds to a difference in $\epsilon NO_3$ of ~0.3.

The model underestimate of nighttime nitrate concentrations during the haze buildup must be because the rate of observed $N_2O_5$ hydrolysis (R2) exceeds even the erroneously high model rate of $NO_2$ aerosol uptake (R4). The haze buildup was characterized by a lower daytime MLH and a deeper nocturnal MLH (inferred from the lack of ozone titration) that resulted in higher nitrate production from $N_2O_5$ hydrolysis (Jordan et al., 2020). The model overly titrates ozone (Fig. 11c) due to insufficient nighttime mixing. We drive additional nocturnal mixing by increasing the sensible heat flux at night from slightly negative (-4 W m$^{-2}$) to weakly positive (+10 W m$^{-2}$), representative of anthropogenic heat fluxes in this region (Hong and Hong, 2016; Varquez et al., 2021). To reduce the rate of R4 from overestimated $NO_2$ and allow for a high rate of R2, we increase the nighttime MLH over land to 300 m as suggested by the observations. This largely resolves the incorrect model ozone titration and the severe model overestimate of nighttime $NO_2$ on 5/23-5/24 and on 5/24-5/25 but does not remedy the early model collapse of the evening mixed layer (Fig. 11). Extending this sensitivity test past the haze buildup results in excess nighttime ozone. This may be due to the increased cloud cover during the haze buildup (Fig. S16), that could cause additional nighttime mixing over average conditions through enhancement of the urban heat island effect (Theeuwes et al., 2019).

Figure 11b shows that increased nighttime mixing allows for $N_2O_5$ hydrolysis (R2) to become the main nighttime pathway for HNO3, with a rate three times greater than $NO_2$ uptake (R4) in the base model. The raised mixed layer height of 300 m prevents this high rate from resulting in overestimated model nitrate. Increased model nighttime nitrate corresponds to an increase in ALWC of 40%. We use the simulations shown in in Fig. 11 to illustrate that model errors in simulating mixed layer dynamics (overly rapid collapse of the evening mixed layer and insufficient nighttime mixing) result in errors in model chemistry. Nighttime measurements of the vertical structure of key species such as ozone, $NO_2$, $N_2O_5$, and $HNO_3$, complemented by sensible heat flux observations, are needed to further constrain model simulations of nighttime nitrate production.

As discussed in Section 4, in addition to the above difficulties in simulating nitrate, the model fails to reproduce observed sulfate during the Transport/Haze period and this corresponds to a 15 µg m$^{-3}$ underestimate in PM2.5 (Table 3). Studies have



shown a strong relationship between increasing RH and conversion of gas-phase precursors to SNA in haze, indicating the occurrence of heterogeneous chemistry in ALWC (Sun et al. Liu et al., 2015; Quan et al., 2015; 2015a; Chen et al., 2016; Wu et al., 2018a). Figure 12 shows the sulfate oxidation ratio, SOR $\equiv \left(\frac{SO_4^{2-}}{SO_2+SO_4^{2-}}\right)$ as a function of RH at Olympic Park and from aircraft observations. In the observations, SOR increases with RH, but this is missing from the model. We take the approach of Wang et al. (2014) and implement heterogeneous uptake of SO₂ on aerosol (not present in the standard model) as a function of RH according to Eq. 4,

$$k_T = \left[\frac{a}{D_g} + \frac{4}{v\gamma}\right]^{-1} ,$$
(4)

where the mass transfer rate ($k_T$) at which a species is lost from the gas-phase is a function of the particle radius ($a$), the molecular diffusion coefficient ($D_g$), the mean molecular speed ($v$), and the reactive uptake coefficient ($\gamma$), or the probability of irreversible reaction. The value for $\gamma$ depends on RH (Wang et al., 2014) according to Eq. 5.

$$\gamma = \gamma_{RH_{50\%}} + \left(\gamma_{RH_{100\%}} - \gamma_{RH_{50\%}}\right)/(100\% - 50\%) \times (RH - 50\%)$$
(5)

The values $\gamma_{RH_{100\%}} = 3 \times 10^{-4}$ and $\gamma_{RH_{50\%}} = 3 \times 10^{-5}$ best fit the observations using the model without the aforementioned adjustments for nitrate simulation. These values are two orders of magnitude slower than in the original formulation of Wang et al. (2014) but similar to more recent studies (Zheng et al., 2015a; Chen et al., 2016). During the Transport/Haze period, this improves model agreement with average surface (Table 3, 15.4 µg m⁻³ vs. 14.7 µg m⁻³) and daytime aircraft (Fig. S17) sulfate observations. Model agreement with daytime aircraft SO₂ observations is degraded, implying that model emissions during the Transport/Haze period are insufficient to produce both the amount of observed SO₂ and sulfate.

During the Transport/Haze period, Choi et al. (2019) estimated a contribution from transported pollution of 68%. However, the inclusion of heterogeneous uptake of SO₂ on aerosol would increase the amount of both locally produced and transported pollution, as the model attributes ~60% of SO₂ to foreign sources and ~40% to local emissions (Fig. S17). We simulate PM₂.₅ with heterogeneous conversion of SO₂ as described above, and then remove South Korean emissions in order to investigate changes to the fraction of transported pollution. Figure 13 shows the model PM₂.₅ composition for each case during the Transport/Haze period, with an additional 15 µg m⁻³ of PM₂.₅ in the model with heterogeneous uptake of SO₂. In the original model, foreign transport accounts for 66% of PM₂.₅ (25 µg m⁻³), but this fraction is reduced to 54% (29 µg m⁻³) in the revised model as the local contribution (13 vs. 24 µg m⁻³) makes up a greater fraction of the increase. Locally produced sulfate increases from only 1% (<1 µg m⁻³) to 25% (6 µg m⁻³) of local PM₂.₅, implying that local SO₂ controls could have an effect on PM₂.₅ levels. Locally produced nitrate increases from 6 µg m⁻³ to 8 µg m⁻³. The total amount of model nitrate (local + foreign) decreases slightly at the surface and aloft (Fig. S17) which we attribute to the impact of sulfate on reducing εNO₃ described above and shown in Fig. S15 but this does not resolve the model nitrate biases described in Section 5.

The previous calculations only account for the missing model sulfate during the Transport/Haze period, and do not account for the incorrect model representation of nighttime nitrate production or overestimated model HNO₃. This accounts for the dramatic increase in ALWC in Fig. 13, which is already overestimated in the original model formulation as shown in Fig 2. Given the uncertainties in revising the model nitrate simulation, we did not assess the policy implications for improving model



nitrate on local vs. transported pollution. A simple test however of the haze buildup with the inclusion of a factor of five increase to $Vd_{HNO3}$, increased nighttime mixing, and the addition of heterogeneous $SO_2$ uptake described above, results in 40% less nitrate and ALWC. As a result, sulfate concentrations are 30% less than in the simulation with heterogeneous $SO_2$ uptake alone. Therefore, studies attempting to determine $\gamma$ to improve sulfate simulations of haze must also consider the impact of model nitrate biases on their parameterization.

## 7 Conclusions

We used aircraft and surface observations from the NIER-NASA KORUS-AQ field campaign in May and June 2016 to evaluate GEOS-Chem simulations of $PM_{2.5}$ composition in the Seoul Metropolitan Area, including during a haze pollution event characterized by high levels of secondary inorganic aerosol. Models generally overestimate nitric acid and the gas-particle partitioning of nitric acid to aerosol and underestimate sulfate during haze events across East Asia (An et al., 2019). This is of concern for using models to determine the fraction of $PM_{2.5}$ pollution that can be controlled using local policy measures in South Korea, and the level to which exceedances of $PM_{2.5}$ standards are caused by long-range transport.

The model underestimated $PM_{2.5}$ in Seoul during the campaign (NMB = -15%) with larger errors in composition. On average, the model underestimated sulfate (-64%) and SOA (-43%) but overestimated nitrate (+36%). Models typically underestimate secondary organic aerosol (SOA, Zhao et al., 2016), and this could be due to missing sources from anthropogenic precursors (Nault et al., 2020). This SOA bias will be investigated in future studies. Aircraft observations, only available during daytime hours, showed model underestimates in sulfate comparable to the bias at the surface. However, modeled nitrate was underestimated aloft, contradicting the model overestimate in the campaign average (which includes nighttime observations). Hourly surface observations showed that this was due to a model overestimate at night. During the campaign, nitrate formation was limited by the supply of nitric acid, which was overestimated against daytime aircraft observations by +100% and contributed to the model nighttime bias. Recent developments to the model wet deposition scheme have significantly improved the simulation of nitrate and nitric acid, but further improvements are unlikely to resolve the model bias.

The model overestimate in nitric acid was not due to overestimated production, insufficient loss to wet deposition, or uptake to dust or seasalt. Increasing the nitric acid dry deposition velocity by a factor of five was required to reconcile the model with observations. Aircraft observations of total nitrate ($TNO_3 = HNO_3$ and $pNO_3$) as a function of photochemical age support this increase. The model underestimate in deposition could be explained by missing treatment of turbulence driven by the urban heat island effect and the heterogeneity of the urban landscape, which would also increase the surface area available for deposition. Here, we only consider the effect on $HNO_3$, but these factors would also impact other species that readily deposit to surfaces such as $NH_3$, which was not measured during the campaign.



Observations of ozone, $NO_2$, and $ClNO_2$ showed that $N_2O_5$ hydrolysis should be the main driver of nighttime nitrate production while the model primarily produced nitrate through aerosol uptake of $NO_2$. The model overly titrated ozone, with an average nighttime concentration of 13 ppb compared to 24 ppb in the observations. This resulted in excess model $NO_2$ and prevented the production of $N_2O_5$. Observations of ozone and of the nighttime mixed layer height implied insufficient nighttime mixing and an overly rapid collapse of the afternoon mixed layer in the model. We attributed these errors to the premature shutdown of afternoon eddies and missing treatment of the urban heat island effect that typically generates a positive nighttime heat flux that is not present in the model. Nighttime measurements of the vertical structure of key species such as ozone, $NO_2$, $N_2O_5$, and $HNO_3$, ideally complemented by surface heat flux observations, are needed to further constrain model nighttime nitrate production, and determine the extent to which the model underestimates nighttime heating and mixing depth.

The model errors in simulating nitrate and nitric acid, mainly arising from overestimated daytime nitric acid and excess nighttime ozone titration, are exacerbated in the simulation of haze pollution. Overestimated nitric acid results in larger values of daytime nitrate during the haze buildup. This could be due to the model underestimate in sulfate as overestimated model pH would allow for increased partitioning of nitric acid to the particle phase. Nighttime nitrate in the model is underestimated during the haze buildup likely due to missing rapid $N_2O_5$ hydrolysis. Sensitivity simulations showed that raising the nighttime mixed layer and providing a positive nighttime sensible heat flux of $+10$ W m$^{-2}$ improved the model simulation of nitrate, ozone, and the $N_2O_5$ pathway for nitrate production during haze. Previous studies have simply raised the nighttime mixed layer and found little effect on simulated pollution (Oak et al., 2019; Miao et al., 2020) but this may be due to missing nocturnal heating from anthropogenic heat release.

The underestimate in model sulfate during the KORUS-AQ haze event is typical of models that often do not include heterogeneous aerosol uptake of $SO_2$ (Wang et al., 2014; Zheng et al., 2015a, 2015b; Shao et al., 2019). Observations of the sulfate oxidation ratio (SOR) as a function of RH supported the need for this pathway as the strong increase in SOR with RH was not present in the model. A simple parameterization of this process increased model sulfate levels from 4 to 15 µg m$^{-3}$ during the haze, in better agreement with observations. However, the success of this parameterization was complicated by model nitrate biases. A simulation of the haze with both improved model nitrate and heterogeneous uptake of $SO_2$ resulted in a 30% reduction in model sulfate over the simulation with heterogeneous uptake of $SO_2$, illustrating the need to consider model biases in sulfate and nitrate simultaneously. GEOS-Chem parameterizations of the urban environment are lacking and cannot be currently adjusted to robustly simulate nitrate during the campaign. However, future studies attempting to simulate sulfate in haze should consider the impact of model nitrate biases on their parameterizations. These studies require models that are able to simulate a large domain to calculate long-range transport but include the detailed parameterizations of the urban environment (urban heat island effect etc.) required to successfully simulate nitrate.

Determining the contribution of local vs. transported PM$_{2.5}$ is essential to the development of successful policy measures to reduce unhealthy pollution levels. Significant effort has gone into this evaluation in South Korea, but with models that have



errors in PM$_{2.5}$ composition (Choi et al., 2019; Kumar et al., 2021). The local PM$_{2.5}$ contribution may be underestimated without including heterogeneous uptake of SO$_2$ on aerosol to produce sulfate during haze. Locally-produced PM$_{2.5}$ increased from 13 to 24 µg m$^{-3}$, decreasing the fraction of foreign pollution from 66% to 54%. Locally-produced sulfate increased from <1 µg m$^{-3}$ to 6 µg m$^{-3}$, implying that controls on SO$_2$ could have a larger impact than in model formulations without this chemistry. As a consequence of the 2013 Clean Air Action plan implemented in China, emissions of inorganic aerosol precursors have been decreasing (Zheng et al., 2018) and concentrations of PM$_{2.5}$ in China have declined by approximately -5 µg m$^{-3}$ per year from 2013-2018 (Zhai et al., 2019). Emission reductions in South Korea may be less rapid (Bae et al., 2021), and thus the impact of long-range transport on future PM$_{2.5}$ pollution events could decline in the future. It is critical for models to improve representations of the interactions between physical processes and chemical production of PM$_{2.5}$ production to support continued local air quality improvements.

### *Code Availability*

The model code used in this work is available at 10.5281/zenodo.5620667.

### *Data Availability*

The KORUS-AQ data archive (KORUS-AQ Science Team, 2019) includes both the aircraft and ground-based measurements from AirKorea, Olympic Park, and KIST. The precipitation data is available at: https://www.ncdc.noaa.gov/cdo-web/datasets. Cloud observations (RKSS ASOS station) are available here: (http://mesonet.agron.iastate.edu/request/download.phtml).

### *Author Contribution*

The original draft preparation was completed by KRT, with review and editing by JHC, BAN, CEJ, HK, and GC. JHC, CEJ, GC, BAN, HK, and KRT contributed to project conceptualization. Modeling work was done by KRT, with additional support from SZ, XW, EM, GL and FY and formal analysis was completed by KRT and BAN. The observational data for this project was provided by BAN, HK, JLJ, PCJ, JED, MJK, SK, IJS, DRB, and LC. JHW and YK provided the KORUS-AQ emissions.

### *Competing Interests*

The authors have the following competing interests: Some authors are members of the editorial board of Atmospheric Chemistry and Physics. The peer-review process was guided by an independent editor, and the authors have also no other competing interests to declare.

### *Acknowledgements*

We acknowledge Gangwoong Lee for his leadership in managing the campaign efforts at Olympic Park. We acknowledge Andrew Weinheimer for the use of his NO and NO$_2$ data from KORUS-AQ. We acknowledge Ron Cohen for the use of the TD-LIF data. We acknowledge Glenn Diskin for the use of DACOM CO and DLH RH data. We acknowledge Bill Brune for



the use of his ATHOS OH data. We acknowledge Paul Wennberg and John Crounse for the use of their CIT-CIMS HNO₃ data. We acknowledge L. Greg Huey for the use of his SO₂ data. We acknowledge James J. Szykman for the use of ceilometer data at Olympic Park. We acknowledge Seogjo Cho for the MARGA data at Olympic Park. We acknowledge Ke Li and Yingying Yan for their help implementing aromatic chemistry in GEOS-Chem. We thank Jerome Fast, Rahul Zaveri, and David Peterson for their helpful discussions. PCJ and JLJ were supported by NASA Grants 80NSSC18K0630 and 80NSSC19K0124. The GEOS-FP data used in this study/project have been provided by the Global Modeling and Assimilation Office (GMAO) at NASA Goddard Space Flight Center.

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



Table 1. Description of the ground site and aircraft observations used in this work[1]

| Instrument | PI | Species | Reference[2] |
|---|---|---|---|
| **Ground Observations** | | | |
| *Korea Institute of Science and Technology (KIST)*[3] | | | |
| Aerodyne High-Resolution Time-of-Flight Aerosol Mass Spectrometer (HR-ToF-AMS) | Hwajin Kim | OA, pNH$_4$, pNO$_3$, pSO$_4$ | Kim et al., 2018 |
| Multi-angle absorption spectrometer (MAAP) | Hwajin Kim | BC | Kim et al., 2018 |
| *Olympic Park*[4] | | | |
| Monitor for AeRosols and Gases in ambient Air (MARGA) | Seogju Cho | SO$_2$, SO$_4^{2-}$ | N/A |
| Chemical Ionization Mass Spectrometry (CIMS) | Saewung Kim | ClNO$_2$ | Slusher et al., 2004 |
| Vaisala CL51 | James Szykman | MLH | N/A |
| 2B Tech 211, Teledyne T200U, Teledyne T500U CAPS, Aerodyne QCL | James Szykman and Andrew Whitehill | O$_3$, NO, NO$_2$ | N/A |
| Dasibi Model 2108 Oxides of Nitrogen Analyzer | NIER | O$_3$, NO$_2$ | N/A |
| BAM-1020 instruments (Met One Instruments, Inc., Grants Pass, OR, USA) | NIER | PM$_{2.5}$ | N/A |
| **DC8 Aircraft** | | | |
| High-Resolution Time-of-Flight Aerosol Mass Spectrometer (HRToF-AMS)[5] | Jose Jimenez | pNO$_3$, pSO$_4$ | Nault et al., 2018 Guo et al., 2021 |
| Soluble Acidic Gases and Aerosol (SAGA) | Jack Dibb | Na$^+$, Cl$^-$ | Dibb et al., 2003 |
| Caltech CIMS (CIT-CIMS) | Paul Wennberg | HNO$_3$, propene hydroxynitrate | St. Clair et al., 2010; Crounse et al., 2006 |
| Airborne Tropospheric Hydrogen Oxides Sensor (ATHOS) | William Brune | OH | Faloona et al., 2004; Brune et al., 2020 |
| NCAR 4-Channel chemiluminescence instrument | Andrew Weinheimer | NO, NO$_2$ | Weinheimer et al., 1993, 1994 |
| Georgia Tech–Chemical Ionization Mass Spectrometer (GT-CIMS) | L. Greg Huey | SO$_2$ | Kim et al., 2007 |
| Diode laser spectrometer (Differential Absorption Carbon monOxide Measurement, DACOM) | Glenn Diskin | CO | Sachse et al., 1987 |
| Diode Laser Hygrometer measurements of H2O(v) (DLH) | Glenn Diskin | RH% | Diskin et al., 2002 |
| Thermal Dissociation–Laser-Induced Fluorescence (TD-LIF) | Ron Cohen | ΣANs, ΣPNs | Wooldridge et al., 2010; Day et al., 2002 |
| Whole Air Sampler (WAS) | Donald Blake | propene | Simpson et al., 2020 |

[1]For a full description of all KORUS-AQ observations, see Crawford et al., 2021.

00    [2]For specific measurement descriptions including uncertainty information, see the KORUS-AQ data archive (doi: 10.5067/Suborbital/KORUSAQ/DATA01)

[3]Korea Institute of Standards and Technology (KIST), 37.602°N, 127.126°E

[4]Olympic Park site in Seoul, 37.522°N,127.124°E

[5]AMS data is written without the charge, see http://cires1.colorado.edu/jimenez-
05    group/wiki/index.php/FAQ_for_AMS_Data_Users#Why_do_you_write_SO4_.26_NO3_and_not_SO42-_.26_NO3-.3F.





Table 2. KORUS-AQ emissions over the domain 70º to 140ºE, 15ºS to 55ºN

| May 2016 (Gg) | NO$_x$ | CO | SO$_2$ | NH$_3$ |
|---|---|---|---|---|
| Natural | 763[1] | NA | 143[3] | 155 |
| Biomass burning | 92 | 7122 | 53 | 137 |
| Fossil fuel combustion[2] | 1921 | 16163 | 2133 | 1705[4] |
| **Total** | 2776 | 23285 | 2329 | 1997 |

[1]Lightning, soil and fertilizer emissions
[2]Point, area, mobile sources, ships, aircraft from the KORUSv5 inventory
10   [3]Volcanic eruptions + degassing
[4]Includes agricultural emissions

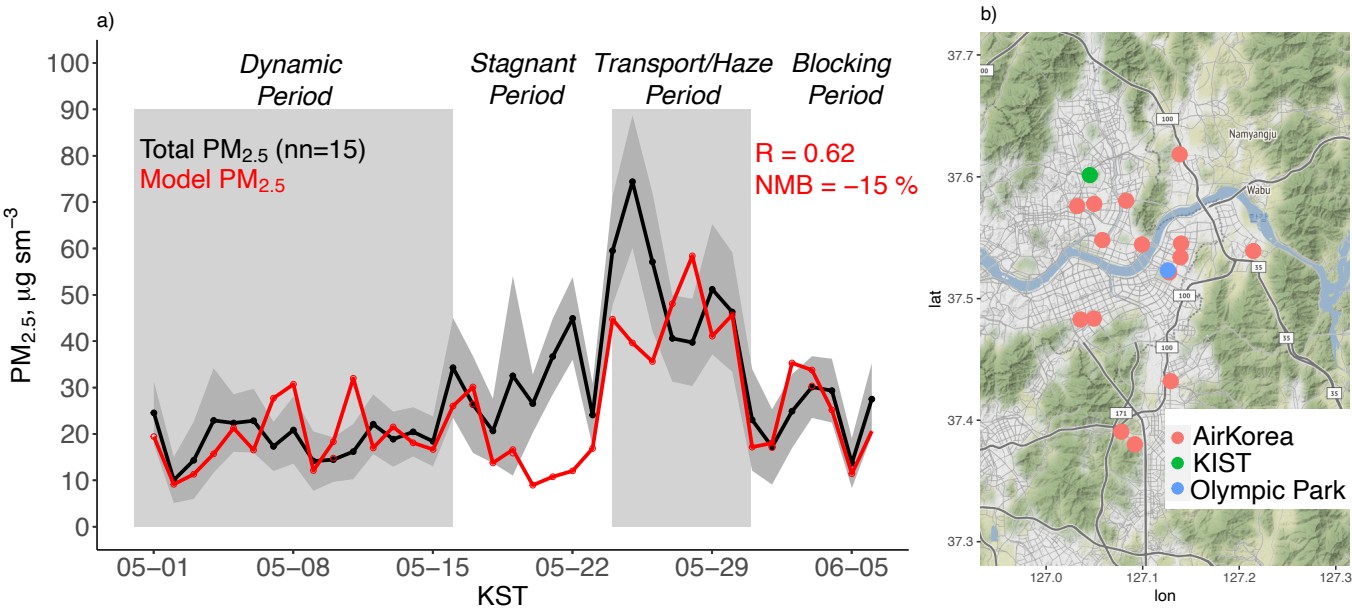

**Figure 1.** a) Model simulation of PM$_{2.5}$ during KORUS-AQ compared against the mean observations at the 15 AirKorea sites
15   in the b) GEOS-Chem model grid-box containing Olympic Park and KIST. The gray shading shows the observed standard
deviation. The correlation coefficient (R) and normalized mean bias (NMB) are inset. Map tiles by Stamen Design, under CC
BY 3.0. Data by © OpenStreetMap contributors, under ODbL.





**Figure 2.** Model simulation of PM$_{2.5}$ compared against observations where the fractional source contributions are calculated from KIST and applied to the mean AirKorea PM$_{2.5}$ observations from Figure 1 during the four meteorological periods. Figure values are shown in Table 3. The radius of each pie chart is scaled to the maximum value of modeled or observed PM$_{2.5}$ (53 µg m$^{-3}$). The blue circles show the aerosol liquid water content (ALWC) associated with PM$_{2.5}$. The sulfate-nitrate-ammonium components are bordered in black to guide the reader.



Table 3. Modeled vs. observed PM$_{2.5}$ composition

| Species | Observations (µg m$^{-3}$) | | | | | Model (µg m$^{-3}$) | | | | |
|---|---|---|---|---|---|---|---|---|---|---|
| | Dynamic | Stagnant | Transport/ Haze | Blocking | Avg | Dynamic | Stagnant | Transport/ Haze | Blocking | Avg |
| Sulfate | 3.9 | 3.6 | 14.7 | 5.5 | 6.1 | 1.7 | 1.4 | 4.1 | 2.1 | 2.2 |
| Nitrate | 2.4 | 3.4 | 11.2 | 3.1 | 4.5 | 4.2 | 4.0 | 12.9 | 6.2 | 6.1 |
| Ammonium | 1.9 | 2.2 | 8.2 | 2.7 | 3.3 | 1.8 | 1.7 | 5.3 | 2.6 | 2.6 |
| SOA | 6.0 | 14.2 | 11.5 | 8.6 | 9.5 | 3.9 | 4.8 | 10.0 | 5.1 | 5.4 |
| POA | 3.3 | 4.3 | 4.8 | 2.8 | 3.7 | 2.8 | 2.8 | 3.3 | 3.3 | 3.0 |
| BC | 1.2 | 1.7 | 2.2 | 1.3 | 1.5 | 1.0 | 1.2 | 2.1 | 1.5 | 1.3 |
| **PM$_{2.5}$** | 18.7 | 29.4 | 52.6 | 24.0 | 28.6 | 15.4 | 15.9 | 37.7 | 20.8 | 20.6 |
| ALWC[1] | 12.0 | 4.1 | 26.9 | 6.2 | 12.6 | 11.9 | 17.6 | 48.7 | 29.5 | 22.9 |
| **PM$_{2.5}$ + H₂O** | 30.7 | 33.5 | 79.5 | 30.2 | 41.2 | 27.3 | 33.5 | 86.4 | 50.3 | 43.5 |

35 [1]Aerosol liquid water content (ALWC) is calculated using E-AIM from temperature at KIST, the 50$^{th}$ percentile of RH across the AirKorea sites in Figure 1b, and the speciated PM$_{2.5}$ components from Figure 2.

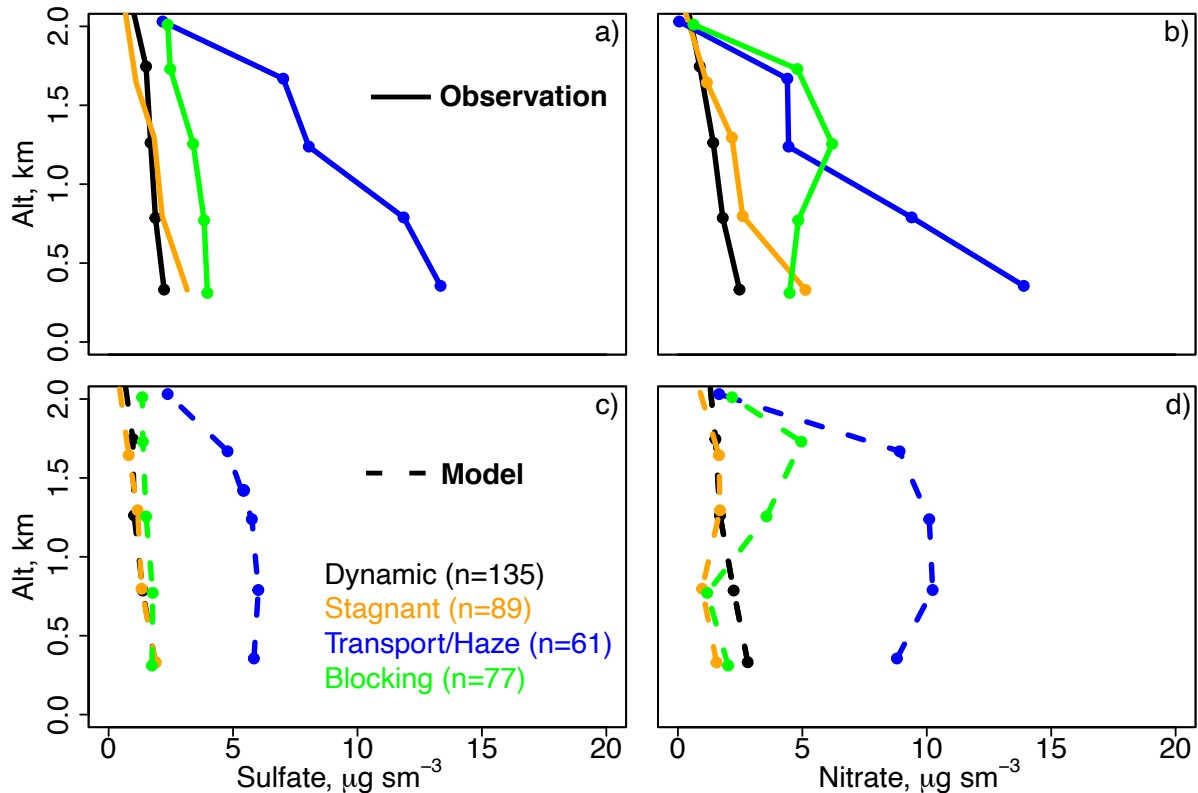

**Figure 3.** Mean vertical profiles of a) observed sulfate, b) observed nitrate, c) model sulfate, and d) model nitrate for the 40 descents into Olympic Park for each meteorological period. The observations (solid lines) and model (dashed lines) are binned to the nearest 0.5 km below 2 km.



**Figure 4.** a) Mean hourly modeled vs. observed nitrate derived from PM₂.₅ observations in the GEOS-Chem gridbox and KIST speciated composition as described in Section 4 for May 1 to June 7, 2016. The gray shading indicates the observed 25$^{th}$ to 75$^{th}$ percentile across the grid box. The model sensitivity studies are described in Section 5. b) Mean model nitric acid diurnal cycle. c) Mean model reactions that produce HNO₃ as described in Section 3.



50

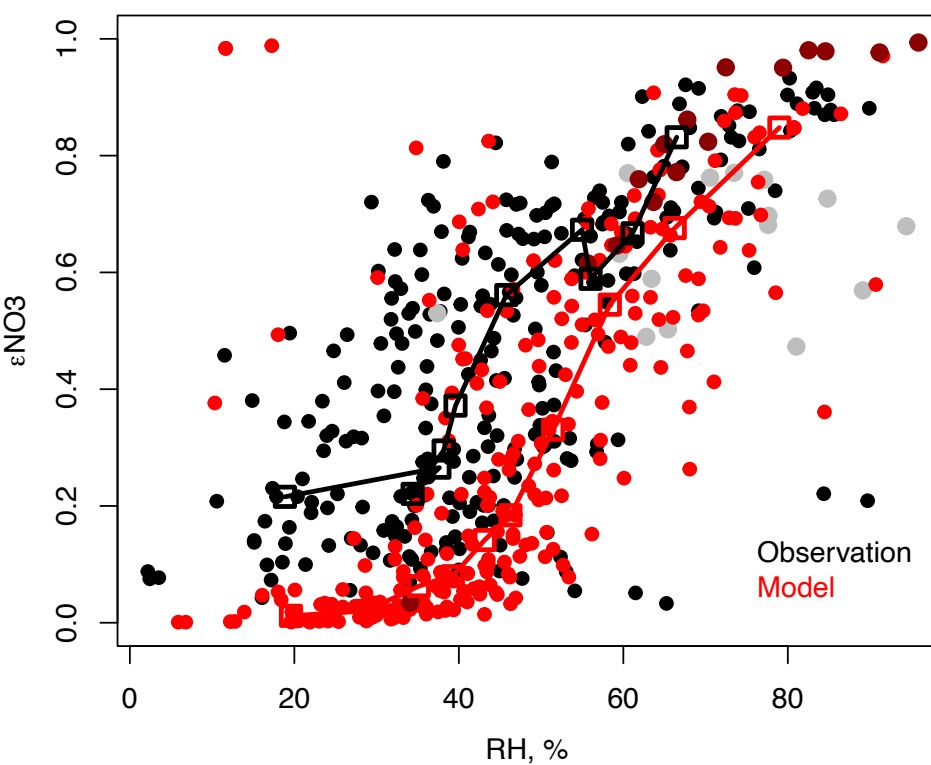

**Figure 5.** Modeled and observed $\varepsilon NO_3$ as a function of RH below 1.5 km for the domain of Fig. 3. Median $\varepsilon NO_3$ as a function of equally size-binned RH is overlaid (squares). The haze buildup (5/24-5/26) is shown in gray for the observations and dark red for the model.

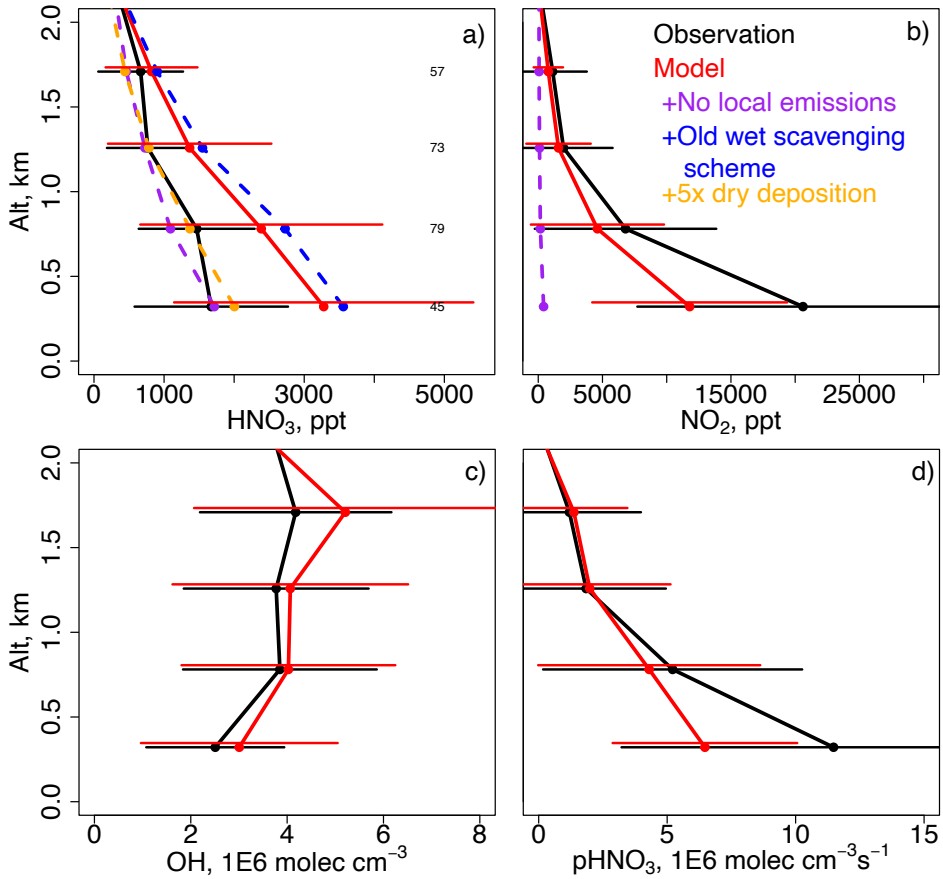

**Figure 6.** Mean vertical profiles of a) HNO₃, b) NO₂, c) OH, and d) production of HNO₃ (pHNO₃) for the same domain as Fig. 3 but accounting for the availability of OH, NO₂, and HNO₃ observations. The horizontal bars show the observed and modeled standard deviations. The number of points in each altitude bin are shown in panel a).

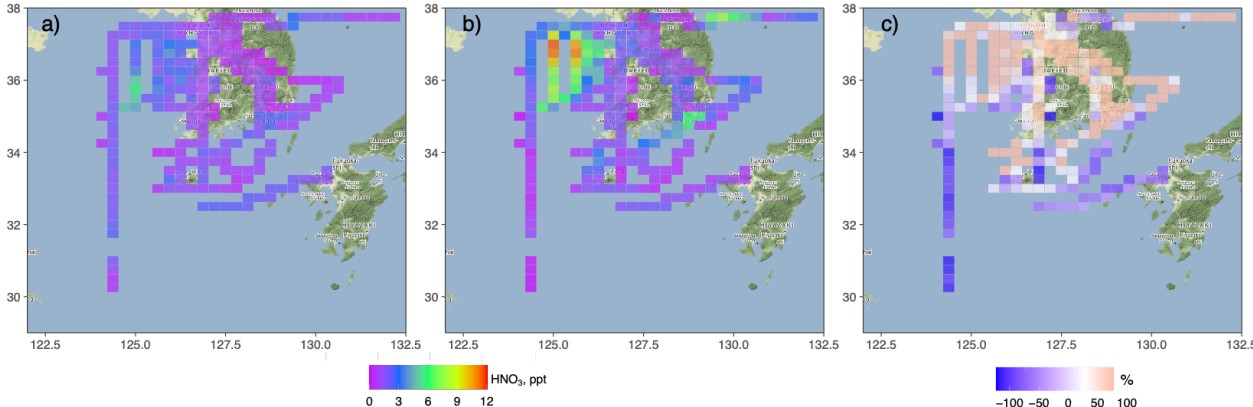

**Figure 7.** Gridded HNO₃ from the observations a), model b), and the percent difference c) along the flight tracks at the model resolution and below 2 km. Map tiles by Stamen Design, under CC BY 3.0. Data by © OpenStreetMap contributors under ODbL.



**Figure 8.** Plot of binned observations of $NO_x$ (right axis), total nitrate ($TNO_3 = HNO_3 + pNO_3$), sum of peroxy nitrates ($\Sigma PNs$), and sum of alkyl- and multi-functional nitrates ($\Sigma ANs$) (left axis for $TNO_3$, $\Sigma PNs$, and $\Sigma ANs$), normalized to background subtracted CO. The background CO from Nault et al. (2018) of 200 ppbv was used. The photochemical age was calculated using propene and one of its photochemical products, propene hydroxynitrate (Section S3). Data are binned between 0 and 5 equivalent hr between 11am to 4pm KST below 1km for the SMA (127 to 127.7°N, 37.2 to 37.7°N). The fit for $NO_x$ (dotted gray curve) is an exponential decay, leading to a first order rate of 0.31 hr⁻¹, which represents the loss of $NO_x$ via the production of oxidized compounds, such as $TNO_3$. The best fit for $TNO_3(t)$ from Eq. 3 (dotted blue curve) includes this production and solves for first order loss, which is assumed to be equivalent to the $TNO_3$ deposition rate (Section S3). Red curves represent solutions for $TNO_3(t)$ from Eq. 3, assuming different deposition velocities ($V_d$) discussed in Section 5.2.



**Figure 9.** Mean diurnal cycle from May 1 to June 7, 2016 for a) ozone and b) $NO_2$ for the AirKorea sites within the GEOS-Chem gridbox (Fig. 1b) and for c) NO and d) $NO_x$ at Olympic Park. The gray shading represents the standard deviation across the AirKorea sites. The solid gray line is the AirKorea site closest to Olympic Park, and the dashed line is the measurement from the EPA (Table 2) at Olympic Park. The sensitivity study (blue line) is described in Section 5.3.

80





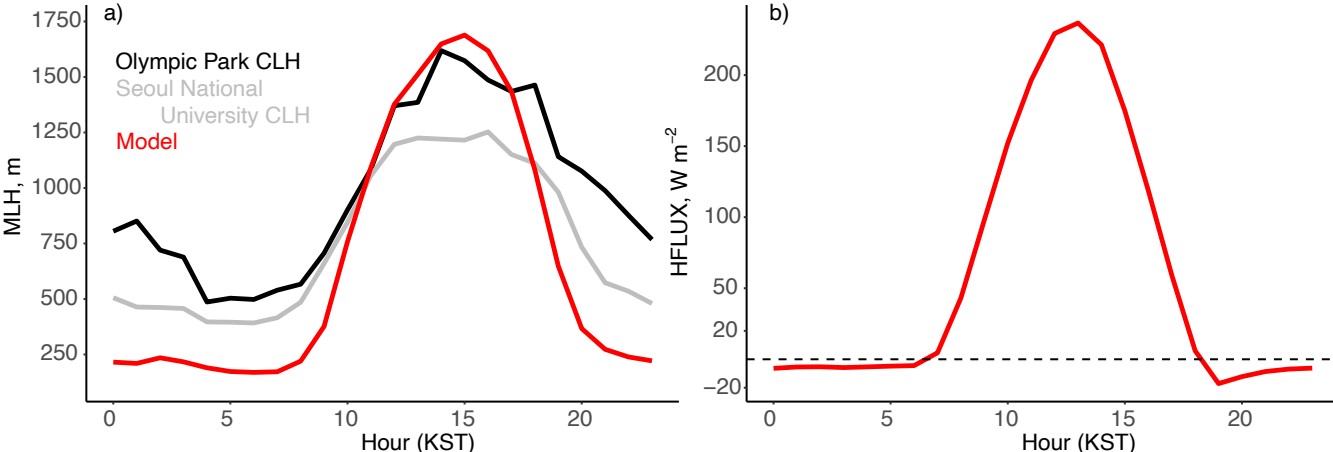

**Figure 10.** a) Mean diurnal cycle for the mixed layer height (MLH) from the model and observations from May 1 to June 7, 2016, and b) sensible heat flux (HFLUX) from the model. The MLH is given for the ceilometers (CLH) at Olympic Park (black) and at Seoul National University (gray).



**Figure 11.** a) Transport/Haze period timeseries of modeled and observed hourly nitrate fraction of PM2.5, b) modeled production of HNO3 from N2O5 (R2) and NO2 (R4), c) ozone, d) and NO2. The sensitivity studies are described in Section 6. The gray shaded regions represent 8pm to 8am.

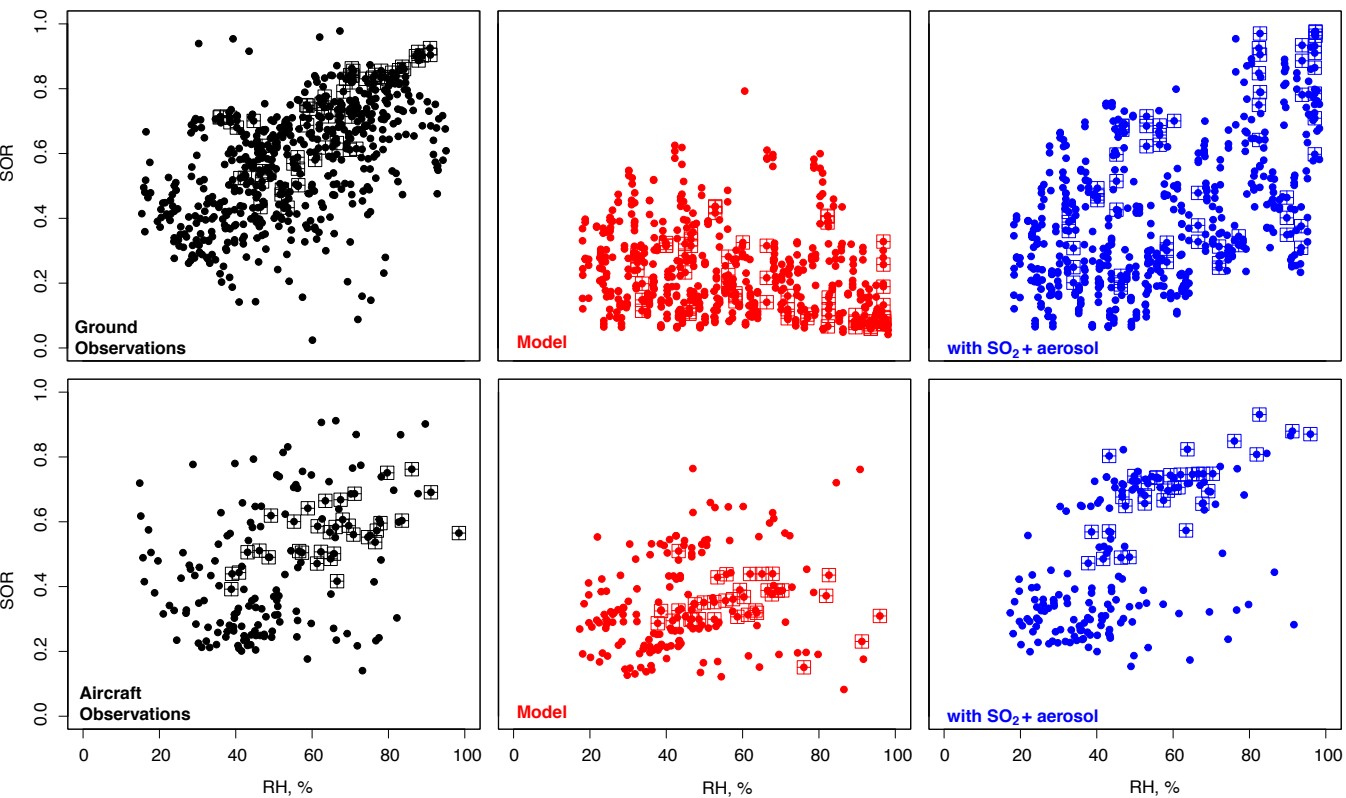

**Figure 12.** Sulfate oxidation ratio (SOR = $\frac{SO_4^{2-}}{SO_4^{2-}+SO_2}$) as a function of RH at Olympic Park and from aircraft below 1km for the same domain as Fig. 3. The squares highlight the data during the Transport/Haze period.





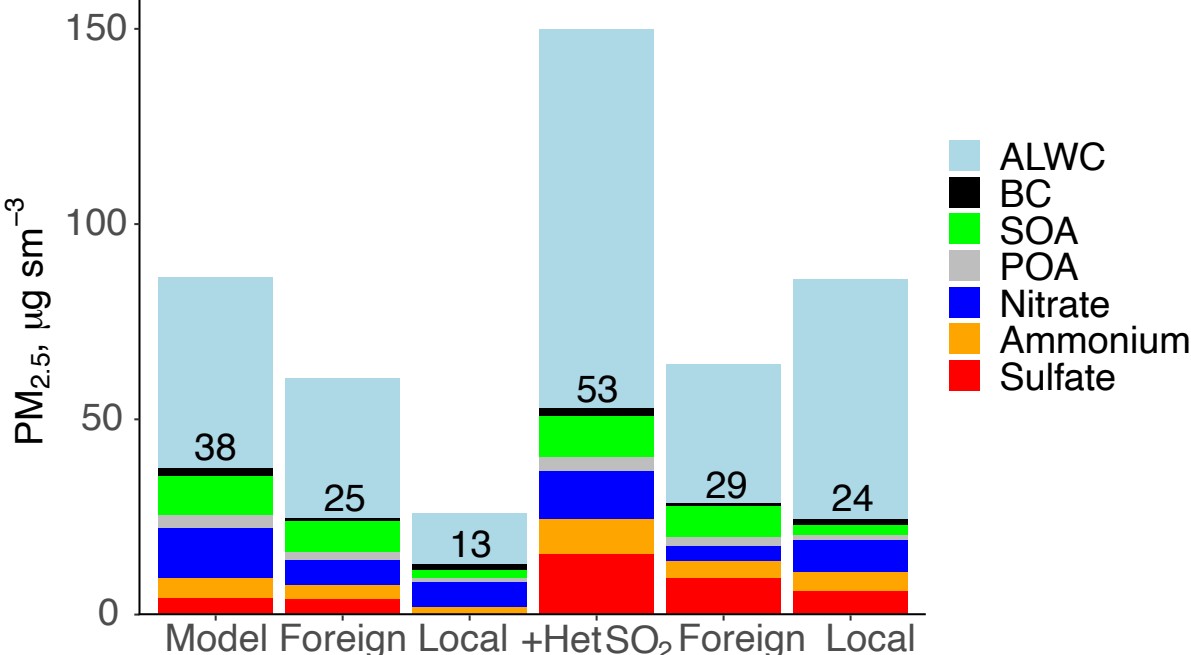

**Figure 13.** Composition of model PM$_{2.5}$ during the Transport/Haze period. The foreign and local contributions and model sensitivity test including heterogeneous uptake of SO$_2$ to aerosol (Het SO$_2$) are calculated as discussed in Section 6. The total PM$_{2.5}$ excluding aerosol liquid water content (ALWC) is given for each simulation.