# Peer review of "Limitations in representation of physical processes prevent successful simulation of PM2.5 during"

_Atmospheric Chemistry and Physics, 2021_

## Referee Comment (RC1)

Review to acp-2021-946: "Limitations in representation of physical processes prevent successful simulation of PM$_{2.5}$ during KORUS-AQ" By Travis et al.

This paper presented a detailed model study of the aerosol composition in Korean during the NASA KORUS-AQ aircraft campaign. The study tried to improve the model simulated PM2.5 and use it to quantify the contribution from long range transport and local emission to the observed PM composition. The topic is relevant to ACP and study is well designed and presented. I suggest ACP publish it with some clarification and improvement.

Science related:

(1) Nitrate, sulfate, and ammonium often show a dynamic and complicated balance. Maybe the analysis can focus more on total sulfate+nitrate+ammonium, where overall model shows better agreement with the observations. Your conclusion on missing local source during the so called 'Transport/Haze' period is probably still hold.

(2) How is the model vs observation comparison for SO2? Only SOR is showed in Figure 12. How is the direct comparison of SO2, esp. in different test runs? SO2 is the primary pollutant and precursor of sulfate, its performance should be relevant.

(3) Heterogenous reaction of SO2 to sulfate has been proposed and studied for long time. Early work should be acknowledged.
Summarized by: Ravishankara, Science, Heterogeneous and Multiphase Chemistry in the Troposphere, 1997.
Initial work: Chameides & Davis, The free radical chemistry of cloud droplets and its impact upon the composition of rain, JGR, https://doi.org/10.1029/JC087iC07p04863, 1982, Application in the CTM: P. Kasibhatla,W. L. Chameides,J. St. John, A three-dimensional global model investigation of seasonal variations in the atmospheric burden of anthropogenic sulfate aerosols, https://doi.org/10.1029/96JD03084, 1997

(4) Based on Figure 11, the 'Increased nighttime mixing' is not necessary improved the model performance systematically. On 05-23 and 05-24 nighttime improvement can been seen for NO2 and ozone. But for other days, the blue line compared worse to observations: phase shift for NO2 on 05-27, 05-28, 05-29 nighttime, and for Ozone on 05-26, 05-27, 05-28, 05-29 'Increased nighttime mixing' degraded the performance.

(5) Please list all model experiments you have done in a table. Quite some model experiments results are discussed, e.g. 'Model', 'Increased nighttime mixing', 'HetSO2', 'No nighttime production', 'Old wet scavenging', '5x dry deposition', etc. What is the difference between those experiments? There is only one thing different between those runs vs the control run, or they are accumulated? 5x dry deposition is global? Or just for urban area or certain particular landuse? Test runs are for both coarse and fine resolution?

(6) Section 5.3 discussion can be shortened and focused on the new findings from this analysis.

As you have shown, even just for the inorganic aerosol components, there are many parameters you can adjust to move some aspect of the model simulation closer to observations. Overall, the model sulfate+nitrate+ammonium show reasonable agreement with observations. If your purpose is to significantly improve the nitrate and sulfate simulation, more evaluations are required, including other region and time.

Model related:
(0) GEOS-Chem has been used in many studies, including multiple aircraft campaigns. It typically shows good performance. It seems the main GEOS-Chem development team is not part of this paper. Have you discussed your findings with the GEOS-Chem development group? 5 times increase of dry deposition and adding a new chemical pathway of SO2 to sulfate seem quite significant changes for a mature model like GEOS-Chem.
(1) Model limitation. This study used a global CTM GEOS-Chem. The model resolution, 0.25x0.3125 is high for a global model, while it might not be the best choice for urban pollution study. Based on Figure 1, most of the ground stations probably are within one grid point of the model. What is the landuse and topography of that model point? What is the reported setting (landuse and elevation) of the stations showed in Figure 1? Do they generally agree? Since GOES-Chem should be able to capture the chemical and physical processes on the regional scale, maybe the model-observation comparison should focus on sites that are representative over the regional conditions?
(2) Model set up: this is a nested simulation, right? From 2x2.5 to 0.25x0.3125? Nesting is a good idea but based on experience with nested model like WRF/WRF-Chem the spatial resolution of ~ 1:3 or 1:4 will provide a smoother and more consistent transition between outer and inner domains. 1:8 is probably too much, for both dynamic and chemical processes.

Other: Figure 12, 'from aircraft below 1km for the same domain as Fig. 3.' Figure 3 didn't show domain info about the aircraft.

---

## Author Comment (AC1)

We thank the two reviewers for their comments and provide our specific responses below in blue with new additions in bold.

**Reviewer #1**

This paper presented a detailed model study of the aerosol composition in Korean during the NASA KORUS-AQ aircraft campaign. The study tried to improve the model simulated PM$_{2.5}$ and use it to quantify the contribution from long range transport and local emission to the observed PM composition. The topic is relevant to ACP and study is well designed and presented. I suggest ACP publish it with some clarification and improvement.

Science related:

- Nitrate, sulfate, and ammonium often show a dynamic and complicated balance. Maybe the analysis can focus more on total sulfate+nitrate+ammonium, where overall model shows better agreement with the observations. Your conclusion on missing local source during the so called 'Transport/Haze' period is probably still hold.

We respectfully disagree with the reviewer that it would be better to focus on sulfate+nitrate+ammonium.

We show the total of all aerosol components in Fig. 1 but have added the following sentence on line 237 to make our argument that it is important to consider components separately.
**"On average, the model simulates SNA within 20%. However, this is due to compensating biases which has implications for controlling precursor species."**

We also discuss model errors in simulating individual components separately as they have different impacts on aerosol properties (i.e. aerosol water, pH etc.) as well as different precursor species. We refer the reviewer to line 239 where we state: "The excess model nitrate is the primary driver of overestimated ALWC (+82%)."

To explain further, a key motivation for the KORUS-AQ campaign was to collect data to help South Korean policy makers improve air quality. That requires an understanding of the primary sources (emissions) and secondary atmospheric processes that produce problematic trace gases and aerosols. Good model agreement with observed PM$_{2.5}$ can be obtained, with discrepancies in its apportionment across the constituents that comprise total PM$_{2.5}$ mass and this has implications for which sources should be controlled. That was the motivation behind this study. To guide policy makers a better understanding of the different precursors, the separate and coupled atmospheric processes driving large accumulations of PM$_{2.5}$ in haze is required.

- How is the model vs observation comparison for SO2? Only SOR is showed in Figure 12. How is the direct comparison of SO2, esp. in different test runs? SO2 is the primary pollutant and precursor of sulfate, its performance should be relevant.

We agree with the reviewer that this is an important comparison. These comparisons were included in the manuscript as described below.

We show the base model performance of SO$_2$ on line 258: "The corresponding profiles for SO$_2$ and nitric acid are shown in Fig. S3."

We show the model performance for sensitivity studies in Fig. S17, starting on line 564. "During the Transport/Haze period, this improves model agreement with sulfate observations at the surface (~15 µg m$^{-3}$ vs. Table 3: 15 µg m$^{-3}$) and aloft (Fig. S17). Model agreement with daytime aircraft SO$_2$ observations is degraded, implying that model emissions during the Transport/Haze period are insufficient to produce both the amount of observed SO$_2$ and sulfate."

- Heterogenous reaction of SO2 to sulfate has been proposed and studied for long time. Early work should be acknowledged.
  - Summarized by: Ravishankara, Science, Heterogeneous and Multiphase Chemistry in the Troposphere, 1997.
  - Initial work: Chameides & Davis, The free radical chemistry of cloud droplets and its impact upon the composition of rain, JGR, https://doi.org/10.1029/JC087iC07p04863, 1982,

- o Application in the CTM: P. Kasibhatla,W. L. Chameides,J. St. John, A three-dimensional global model investigation of seasonal variations in the atmospheric burden of anthropogenic sulfate aerosols, https://doi.org/10.1029/96JD03084, 1997

We thank the reviewer for these citations. The third citation is most relevant to this work, and has been added on line 105, **"Early modeling work suggested that this chemistry must be occurring generally in the polluted boundary layer in the United States and Europe (Kasibhatla et al., 1997)."**

- ▪ Based on Figure 11, the 'Increased nighttime mixing' is not necessary improved the model performance systematically. On 05-23 and 05-24 nighttime improvement can been seen for NO2 and ozone. But for other days, the blue line compared worse to observations: phase shift for NO2 on 05-27, 05-28, 05-29 nighttime, and for Ozone on 05-26, 05-27, 05-28, 05-29 'Increased nighttime mixing' degraded the performance.

We agree with the reviewer that the "increased nighttime mixing" does not systematically improve the model performance. This was acknowledged in the paper and we refer the reviewer back to line 533:

"Extending this sensitivity test past the haze buildup results in excess nighttime ozone. This may be due to the increased cloud cover during the haze buildup (Fig. S16), that could cause additional nighttime mixing over average conditions through enhancement of the urban heat island effect (Theeuwes et al., 2019)."

- ▪ Please list all model experiments you have done in a table. Quite some model experiments results are discussed, e.g. 'Model', 'Increased nighttime mixing', 'HetSO2', 'No nighttime production', 'Old wet scavenging', '5x dry deposition', etc. What is the difference between those experiments? There is only one thing different between those runs vs the control run, or they are accumulated? 5x dry deposition is global? Or just for urban area or certain particular landuse? Test runs are for both coarse and fine resolution?

Thank you for this suggestion. Table 4 has been added describing the sensitivity simulations in the main text.

- ▪ Section 5.3 discussion can be shortened and focused on the new findings from this analysis.

We have removed the discussion about Figure S12 and the associated sensitivity study and put in in the supplement (lines 486-493 are now Section S6). We have replaced that discussion with this line starting on 482 to refer to the supplement: **"We illustrate in Section S6 that reducing the collapse of the evening MLH without a change to the drivers of mixing (i.e., heat fluxes, friction velocity) has negligible impact on decreasing model ozone titration (Fig. S12)."**

To further address the reviewers concern about length, we have moved the paragraph on lines 220-231 to the supplement (Section S2).

As you have shown, even just for the inorganic aerosol components, there are many parameters you can adjust to move some aspect of the model simulation closer to observations. Overall, the model sulfate+nitrate+ammonium show reasonable agreement with observations. If your purpose is to significantly improve the nitrate and sulfate simulation, more evaluations are required, including other region and time.

We agree with the reviewers that further evaluations in other time periods are needed, and this paper lays the groundwork for those studies. We added the following sentences on line 593,

**"Follow-up work will include consideration of improvements to the model sulfate and nitrate simulation with a coupled model system such as WRF-GC (Lin et al., 2020) that is able to better simulate the urban scale as well as long-range transport."**

and line 668:

**"Follow-up studies to this work will evaluate model performance during other seasons (i.e., winter) using a model system with online meteorology to determine whether factors driving model errors in this work are occurring throughout the year."**

Model related:

- GEOS-Chem has been used in many studies, including multiple aircraft campaigns. It typically shows good performance. It seems the main GEOS-Chem development team is not part of this paper. Have you discussed your findings with the GEOS-Chem development group? 5 times increase of dry deposition and adding a new chemical pathway of SO2 to sulfate seem quite significant changes for a mature model like GEOS-Chem.

GEOS-Chem is a model that undergoes continuous development (https://geos-chem.seas.harvard.edu/geos-new-developments) and encourages community input. The lead author and several of the co-authors have contributed to those developments, including Xuan Wang, Erin McDuffie, and Shixian Zhang. The primary author trained with the GEOS-Chem model scientist as her dissertation advisor.

We respectfully disagree that the GEOS-Chem model typically shows good performance. We refer the reviewer to our discussion of the other GEOS-Chem papers that show similar issues to what we show here: overestimated $HNO_3$ and underestimated sulfate during haze.

On line 353: "Previous attempts to improve model nitrate invoked an unknown sink of $HNO_3$ in the model (Heald et al., 2012; Weagle et al., 2018), as uncertainties in precursor emissions, the rate of $N_2O_5$ hydrolysis (R2/R3) or gas-phase production (R1), OH concentrations, and $HNO_3$ dry deposition velocity ($Vd_{HNO3}$) could not explain model nitrate biases."

On line 501: "The failure of models to simulate sulfate production in haze in East Asia is a current topic of intensive research and is attributable to missing sulfate production in aerosol water (Wang et al., 2014; Zheng et al., 2015a; Chen et al., 2016; Shao et al., 2019; Miao et al., 2020)."

- Model limitation. This study used a global CTM GEOS-Chem. The model resolution, 0.25x0.3125 is high for a global model, while it might not be the best choice for urban pollution study. Based on Figure 1, most of the ground stations probably are within one grid point of the model. What is the landuse and topography of that model point? What is the reported setting (landuse and elevation) of the stations showed in Figure 1? Do they generally agree? Since GOES-Chem should be able to capture the chemical and physical processes on the regional scale, maybe the model-observation comparison should focus on sites that are representative over the regional conditions?

We agree with the reviewers that the ground stations are within one grid point of the model. In the paper, we average all the ground stations in Figure 1 for comparison to the model. This is stated on line 213, "Figure 1a shows the model simulation of daily average $PM_{2.5}$ (Eq. 1) compared to the observed average of the 15 AirKorea sites within the GEOS-Chem grid box containing the major SMA monitoring sites (KIST and Olympic Park)."

We respectfully disagree that the model resolution is insufficient for an urban pollution study, as the motivation for this work is to better understand the impact of long-range transport which requires a global CTM. In the paper, we state the following on line 77: "Quantifying the effect of long-range transport relies on regional to global-scale models that trade-off the high resolution needed to resolve urban scales with a large enough domain to represent both the study area and upwind source regions."

To address the reviewer's question about land-use, we add the following sentence on line 376 to clarify our argument that the model poorly treats the urban landcover characteristics: **"The model surface roughness, an important parameter governing turbulence, is just 0.1 m in Seoul, compared to values measured between 1 and 3 for forested or urban parts of the city (Hong and Hong, 2016)."**

- Model set up: this is a nested simulation, right? From 2x2.5 to 0.25x0.3125? Nesting is a good idea but based on experience with nested model like WRF/WRF-Chem the spatial resolution of ~ 1:3 or 1:4 will provide a smoother and more consistent transition between outer and inner domains. 1:8 is probably too much, for both dynamic and chemical processes.

We respectfully disagree with the reviewers that the nesting setup is inappropriate. GEOS-Chem uses assimilated meteorology, thus dynamical processes are pre-calculated by the GEOS-FP system. Nesting in GEOS-Chem from 2x25 to 0.25x0.3125 is standard practice. We state in the paper on line 143: "The model is driven by assimilated meteorological data from the NASA Global Modeling and Assimilation Office (GMAO) Goddard Earth Observing System Forward-Processing (GEOS-FP) atmospheric data assimilation system. GEOS-FP has a native horizontal

resolution of 0.25º × 0.3125º, which we apply with the nested version of GEOS-Chem (Chen et al., 2009) over East Asia (70° - 140°E, 15°S - 55°N) using boundary conditions from a global simulation at 2.0° × 2.5° with a 1-month initialization period."

- ▪ Other: Figure 12, 'from aircraft below 1km for the same domain as Fig. 3.' Figure 3 didn't show domain info about the aircraft.

We revised to **"for the descents over Olympic Park"**.

**Reviewer #2**

This is a very comprehensive study elucidating the causes of model biases in PM$_{2.5}$ inorganic components simulated by the GEOS-Chem chemical transport model, using observations from the Korean United States-Air Quality (KORUS-AQ) field campaign. Model deficiencies, including overestimates of daytime HNO$_3$ and nighttime nitrate, too rapid uptake of NO$_2$ by aerosols at night, and underestimates of sulfate during pollution events, were identified and analyzed. A series of model sensitivity simulations were carefully designed in order to lower the model biases and to interpret the possible causes of these model biases.

The study is well conducted and presented, and is an important piece of work advancing our understanding on the processes affecting the inorganic aerosol simulation. I thus recommend publish on ACP. Below are several comments for clarification in the manuscript.

Specific comments:

1) Page 7, Line 25,
The linkage of excess model nitrate with overestimated ALWC (aerosol liquid water content) needs some explanation. How ALWC is estimated in the GEOS-Chem model and in the observations? Why overestimated nitrate lead to higher ALWC, although model PM2.5 is biased low? Please clarify.
We refer the reviewer to our description of the ALWC calculation on line 232. "Figure 2 and Table 3 include the ALWC associated with PM$_{2.5}$, calculated for the observations using the E-AIM IV thermodynamic model (Clegg and Brimblecombe, 1990; Clegg et al., 1998; Massucci et al., 1999; Wexler and Clegg, 2002; Nault et al., 2021b), and ISORROPIAv2.2 (Pye et al., 2009) in GEOS-Chem.

The model is biased low in sulfate and OA but high in nitrate, as discussed on line 238: "The model underestimated sulfate (-64%), overestimated nitrate (+36%), and underestimated SOA (-43%)". Thus PM$_{2.5}$ is less biased due to compensating errors. We also refer the reviewer to the statement on line 239 that "The excess model nitrate is the primary driver of overestimated ALWC (+82%)." See also the line added to address Reviewer #1's comment on line 237:
**"On average, the model simulates SNA within 20%. However, this is due to compensating biases which has implications for controlling precursor species."**

2) Page 10, 2nd paragraph,
As shown in Figure 5, the model epsilon-NO3 values are overall biased low suggesting excess partitioning to the gas phase. Are there any sensitivity simulations conducted in the study that can improve the simulated NO3 gas-aerosol partitioning? How about the one with enhanced dry deposition velocities?
We thank the reviewer for asking this question. We incorrectly stated that the error in model partitioning could be due to overestimated HNO$_3$. We have removed the following sentences starting on line 317: "This low bias in εNO$_3$ could be due to overestimated HNO$_3$, as the lower RH and associated higher temperatures generally prevent excess HNO$_3$ (denominator of Eq. 2) from partitioning to the aerosol-phase. We discuss the possibility of overestimated model HNO$_3$ below."

We added the following text on line 317 to address the reviewer's comment about what would improve model partitioning:
**"This could be a result of underestimated ammonia, not measured during the campaign, or errors in model temperature and RH."**

3) Page 16, 2nd paragragh,
It is not clear how the model treated the simulation with increased sensible heat flux as GEOS-Chem used assimilated meteorology. Which processes and parameters would be affected in this simulation? Would the other sensitivity simulation with increased PBL have the same effect?

We have added the following text on line 535 as clarification **"As the meteorology in GEOS-Chem is calculated offline (Section 3), increasing surface sensible heat flux only impacts the boundary layer mixing parameterization but not the simulation of other meteorological fields. Future work should use a coupled system to investigate other effects of the urban heat island effect on air quality."**

4) Page 36, Figure 6,
The panels c and d of Figure 6 did not show the simulated results from the other three simulations, e.g., 5x dry deposition. Was there any reason?
We thank the reviewer for pointing out this potentially confusing omission. We revised Figure 6 to show all sensitivity simulations for all panels, and add the following text to the caption of Figure 6 – **"Model sensitivity simulations that are not significantly different than the base model run are plotted underneath the base model line."**

5) 9th line of Page 11, 4th line of Page 15, Here "Section 2" should be "Section 3".
Fixed.

**Other revisions**

We have made edits for language clarity throughout the paper to further address Reviewer #1's concern about length, and revised the following numbers:

Line 59 – We intended to reference the fraction of local sulfate in local PM2.5 (25%), but incorrectly wrote the fraction of local sulfate in total sulfate (46%).

Table 2 – We found a rounding error in our calculations that revises the fossil fuel NOx down by 1 Gg but has no impact on any paper statements or conclusions.

Line 264: We were using relative humidity observations from the wrong instrument. The revised RH numbers are now using the DLH relative humidity. As a result we have removed the following sentence which is no longer valid: "If the model RH simulation was unbiased, we would expect an improved simulation of nitrate as the minimal RH bias during the Dynamic period corresponds to the best nitrate simulation (Fig. 3, Fig. S3)."

Line 299: We were using inconsistent definitions of nighttime and daytime for these numbers. The revised temperature and RH values are now using the definitions given in the text (6am and 6pm KST (daytime), 6pm to 6am KST (nighttime).

Figure 11 – We edited the gray shading to have a consistent nighttime (6pm to 6am KST) definition with the text.

Line 442: We found a small error in the calculation of these ozone numbers, they have been revised but have no impact on the conclusions.

Figure 12 – We edited the inset sensitivity study label to be consistent with Table 4.

---

## Referee Report (RR1)

I suggest that ACP publish this paper with three minor modifications/clarifications.

(1) Missing heterogenous chemistry of $SO_2$ is listed as the main reason for model underestimate of sulfate.  But the model result of adding $SO_2$ heterogenous chemistry is only briefly mentioned and the figure is in the supplementary section S17.  I suggest incorporating it in the main part.

(2) In the abstract, the shallow nighttime PBL height, or rapid collapse of mixing layer, is listed as one of main reasons for model bias in nighttime chemistry.  But based on Figure 9, the run with rising PBL (to the observation level) shows no improvement in model performance, i.e. the red line and blue line are almost the same for most species. Please clarify.

(3) Figure 6, where is 5x dry deposition run results in (b), (c), and (d)?  No impact on other species?  $NO_2$ deposition is under-estimated?  Increasing or decreasing dry deposition look more like tuning model toward observations.

One more suggestion: the "No local emissions" run can be a very useful model simulation to quantify the relative contribution from the local emissions vs long range transport. It is not showed in most analyses (figures), right?

The line number in the draft is only partially labelled and mismatched to the number mentioned in the response to reviewer.  We have to use search function to locate the content mentioned in 'ResponsetoReviewers'.

---

## Author Response (AR2)

We thank both anonymous reviewers for their careful read of the revised manuscript. Our responses are below in blue with revisions in bold.

**Reviewer #1**

(1) Missing heterogenous chemistry of SO2 is listed as the main reason for model underestimate of sulfate. But the model result of adding SO2 heterogenous chemistry is only briefly mentioned and the figure is in the supplementary section S17. I suggest incorporating it in the main part.

Thank you for this suggestion. We agree this would be helpful. We have moved Figure S17 to the main text as Figure 13. We added the following to line 522 to further describe the impact of this figure in the main text.

"The derived values for $\gamma$ described above may need to be revised in future work to consider the impacts of errors in the nitrate simulation (discussed below) as well as errors in $SO_2$ emissions."

(2) In the abstract, the shallow nighttime PBL height, or rapid collapse of mixing layer, is listed as one of main reasons for model bias in nighttime chemistry. But based on Figure 9, the run with rising PBL (to the observation level) shows no improvement in model performance, i.e. the red line and blue line are almost the same for most species. Please clarify.

Restate abstract

As the model is run with assimilated meteorology, artificially raising the PBL is not coupled to a physical cause (i.e. increased heat fluxes), hence the effect of changing the height is limited. We explain this with respect to ozone on line 448 and add the word "significant" to add clarity:

"We illustrate in Section S6 that reducing the collapse of the evening MLH without a **significant** change to the drivers of mixing (i.e., heat fluxes, friction velocity) also has negligible impact on decreasing model ozone titration (Fig. S12b)."

On line 481, we discuss the need to address the drivers of mixing (i.e., not just the outcome of an increased PBL height) and add a reference to Table 4 to increase clarity:

"We drive additional nocturnal mixing **(Table 4, increased nighttime mixing)** by increasing the sensible heat flux at night from slightly negative (-4 W m$^{-2}$) to weakly positive (+10 W m$^{-2}$), representative of anthropogenic heat fluxes in this region (Hong and Hong, 2016; Varquez et al., 2021)."

Line 482 explains the result of this sensitivity test:

"This sensitivity test (Table 4) largely resolves the incorrect model ozone titration and the severe model overestimate of nighttime $NO_2$ on 5/23-5/24 and on 5/24-5/25…"

(3) Figure 6, where is 5x dry deposition run results in (b), (c), and (d)? No impact on other species? NO2 deposition is under-estimated? Increasing or decreasing dry deposition look more like tuning model toward observations.

The model runs are plotted underneath the base model run, thus they are difficult to see. We refer the reviewer to this statement in our caption of Figure 6: Model sensitivity simulations that are not significantly different than the base model run are plotted underneath the base model line.

We have tuned HNO3 dry deposition towards observations but are clear that our analysis shows that this tuning is suggestive of the need for stronger loss, and here we implement that using dry deposition. See line 567:

"The model overestimate in nitric acid was not due to overestimated production, insufficient loss to wet deposition, or uptake to dust or seasalt. Increasing the loss of nitric acid, implemented here as an increase in the nitric acid dry deposition velocity by a factor of five, was required to reconcile the model with observations."

- One more suggestion: the "No local emissions" run can be a very useful model simulation to quantify the relative contribution from the local emissions vs long range transport. It is not showed in most analyses (figures), right?

We agree that this sensitivity test is essential to show the relative contribution of local emissions vs. long-range transport. This sensitivity test is used for this purpose in Fig. 13 to calculate the foreign contribution to $PM_{2.5}$. This may not have been clear, so we added the following reference to Table 4 to line 528:

 We simulate $PM_{2.5}$ with heterogeneous conversion of $SO_2$ as described above, and then remove South Korean emissions **(Table 4),** in order to investigate changes to the fraction of transported pollution.

We also realized that a sensitivity test was missing from Table 4 where we remove local emissions with the addition of heterogeneous $SO_2$ chemistry.  This has been added.

Table 4. Description of model experiments

| Name | Resolution | Simulation Length | Description of changes |
|---|---|---|---|
| Base model | $0.25^{o} \times 0.3125^{o}$ over East Asia. Boundary conditions (BCs) from a global $2^{o} \times 2.5^{o}$ simulation[1.] | 1 month initialization + KORUS-AQ period (May 1-June 9). | N/A |
| No nighttime production | $0.25^{o} \times 0.3125^{o}$ over East Asia. | KORUS-AQ period | Remove reactions R2-R5. |
| Old wet scavenging scheme | $0.25^{o} \times 0.3125^{o}$ over East Asia. | KORUS-AQ period | Remove recently implemented wet scavenging scheme (Luo et al., 2019). |
| 5x dry deposition | $0.25^{o} \times 0.3125^{o}$ over East Asia. | KORUS-AQ period | Increase the deposition velocity of $HNO_3$ by a factor of 5. |
| No local emissions | $0.25^{o} \times 0.3125^{o}$ over East Asia. | KORUS-AQ period | Turn off anthropogenic emissions over South Korea. |
| Raise nighttime PBL | $0.25^{o} \times 0.3125^{o}$ over East Asia. | KORUS-AQ period | Increase the nighttime MLH to 500m. |
| Increased nighttime mixing | $0.25^{o} \times 0.3125^{o}$ over East Asia. | May 23 to May 31 | Increase the nighttime MLH to 300m and set nighttime sensible heat flux to 10 W m$^{-2}$. |
| Het $SO_2$ | $0.25^{o} \times 0.3125^{o}$ over East Asia. | KORUS-AQ period | Uptake of $SO_2$ on aerosol with $\gamma_{RH_{100\%}} = 3 \times 10^{-4}$ and $\gamma_{RH_{50\%}} = 3 \times 10^{-5}$. |
| **Het $SO_2$ with no local emissions** | **$0.25^{o} \times 0.3125^{o}$ over East Asia.** | **KORUS-AQ period** | **Uptake of $SO_2$ on aerosol with $\gamma_{RH_{100\%}} = 3 \times 10^{-4}$ and $\gamma_{RH_{50\%}} = 3 \times 10^{-5}$ and turn off anthropogenic emissions over South Korea.** |

[1]Boundary conditions from the base simulation are applied to all sensitivity simulations.

- The line number in the draft is only partially labelled and mismatched to the number mentioned in the response to reviewer. We have to use search function to locate the content mentioned in 'ResponsetoReviewers'.

We apologize for this confusion, we assume this was some error in the upload.

**Reviewer #2**

After reading the response letter and the revised manuscript, I think the authors have adequately addressed both reviewers' comments, and suggest publish on ACP.

Minor typo: Table 4, Base model resolution, '2 x 2.25' should be '2 x 2.5'

This has been fixed.